# PatchDNA: A Flexible and Biologically-Informed alternative to Tokenization for DNA

**Alice Del Vecchio, Chantriolnt-Andreas Kapourani, Abdullah M Athar, Agnieszka Dobrowolska, Andrew Anighoro**, **Benjamin Tenmann**, **Lindsay Edwards** and **Cristian Regep**
Relation Therapeutics
{alice,andreas,abdullah,aga,andrew,ben,lindsay,cristian}@relationrx.com

## Abstract

DNA language models are emerging as powerful tools for representing genomic sequences, with recent progress driven by self-supervised learning. However, performance on downstream tasks is sensitive to tokenization strategies reflecting the complex encodings in DNA, where both regulatory elements and single-nucleotide changes can be functionally significant. Yet existing models are fixed to their initial tokenization strategy; single-nucleotide encodings result in long sequences that challenge transformer architectures, while fixed multi-nucleotide schemes like byte pair encoding struggle with character level modeling. Drawing inspiration from the Byte Latent Transformer's combining of bytes into patches, we propose that 'patching' provides a competitive and more efficient alternative to tokenization for DNA sequences. Furthermore, patching eliminates the need for a fixed vocabulary, which offers unique advantages to DNA. Leveraging this, we propose a biologically informed strategy, using evolutionary conservation scores as a guide for 'patch' boundaries. By prioritizing conserved regions, our approach directs computational resources to the most functionally relevant parts of the DNA sequence. We show that models up to an order of magnitude smaller surpass current state-of-the-art performance in existing DNA benchmarks. Importantly, our approach provides the flexibility to change patching without retraining, overcoming a fundamental limitation of current tokenization methods.

## 1 Introduction

Self-supervised learning has led to a surge of interest in DNA language models, sequence models trained on raw nucleotide data to produce general-purpose genomic representations. These models have shown promise across diverse tasks, from identifying regulatory elements to variant effect prediction (Brixi et al., 2025; Nguyen et al., 2023; Schiff et al., 2024). A central challenge in adapting language modeling to DNA is how to tokenize the input sequence. Unlike natural language, where subword or word-level tokenization can exploit semantic structure and redundancy (Mielke et al., 2021), genomic sequences encode both fine-grained (e.g. letter level single-nucleotide variants) and coarse-grained (regulatory elements) information, often within the same genomic region. The choice of tokenization thus directly impacts both resolution and efficiency.

Existing DNA models typically fix their tokenization strategy prior to training. Models that operate at the single-nucleotide level preserve maximal resolution but produce extremely long sequences that challenge transformer architectures. Conversely, fixed multi-nucleotide schemes such as k-mers or byte pair encoding improve efficiency but often lose critical single-base information. Prior work has shown that downstream performance can be highly sensitive to this tradeoff (Lindsey et al., 2025; Patel et al., 2024). Therefore exploring alternative tokenization strategies and their suitability for encoding DNA sequences is a compelling research direction.

The Byte Latent Transformer (BLT), originally proposed for natural language processing, introduces a dynamic alternative to tokenization, that segments input sequences into variable-length patches based on predictive entropy (Pagnoni et al., 2025). This enables models to allocate at-

tention and computation to regions of high uncertainty, capturing context-dependent structure more effectively. Recognizing the potential advantages that patching offers for genomic data, we introduce `PatchDNA`, a model that represents DNA sequences as contiguous, dynamically determined patches rather than individual tokens (Figure 1). This general framework aligns naturally with the structure of genomic data and offers clear advantages over traditional tokenization for DNA. Patching improves efficiency as patch sizes can far exceed the size of tokens, while preserving single-nucleotide resolution. Moreover, the lack of fixed vocabulary also offers greater flexibility than tokenization, enabling the design of more biologically informed approaches.

**Our key contributions can be summarized as follows:**

- We extend dynamic patching for DNA by modifying the BLT framework and show that patches are a better alternative to token-level representations of genomic sequences in efficiency and flexibility.

- We introduce a novel conservation-guided patching scheme that leverages evolutionary signals to guide patch boundaries, providing a biologically informed inductive bias.

- We introduce re-patching, allowing the patching strategy of the model to be changed after pretraining, overcoming a fundamental limitation of current tokenization methods. This enables flexible downstream application with minimal computational overhead.

Through extensive experiments, we show that conservation-guided patching systematically achieves the strongest results, while alternative patching strategies remain competitive. We further demonstrate the flexibility of the framework through re-patching, enabling models to adapt their patching strategy for different downstream tasks with no retraining from scratch. Our results demonstrate the value of patching in advancing genomic language modeling.

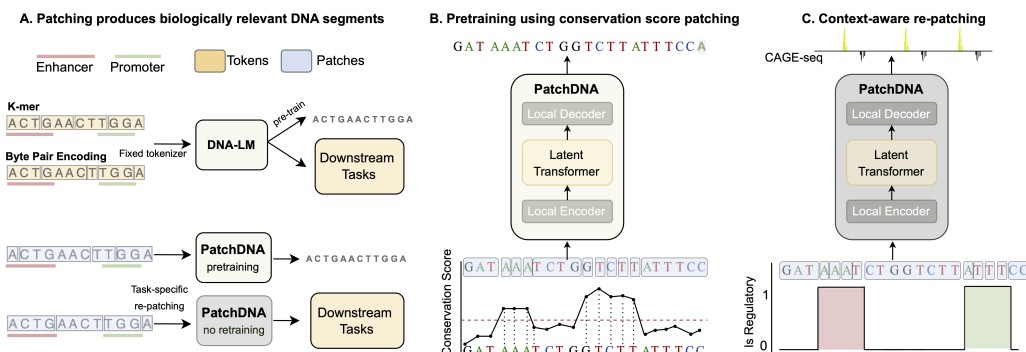

Figure 1: Overview of PatchDNA. (**A**) Unlike fixed tokenization methods, PatchDNA segments sequences into biologically meaningful patches without relying on a fixed vocabulary. (**B**) During pretraining, patch boundaries are guided by evolutionary conservation scores, enabling the model to focus computational resources on functionally important regions. (**C**) We introduce re-patching, enabling flexible downstream application with no retraining from scratch.

## 2 EXISTING DNA TOKENIZATION SCHEMES

Several tokenization strategies offer trade-offs between vocabulary size, biological interpretability, computational efficiency and exhaustiveness of coverage:

**K-mers:** The input sequence is split into fixed-length sub-strings of length $k$, as done in the Nucleotide Transformer (Dalla-Torre et al., 2025). However, small changes to the input sequence can drastically alter the tokenized sequence, making it difficult for the model to align representations of near-identical inputs. This inconsistency hinders efficient learning and may degrade model performance (Zhou et al., 2024).

**Byte-Pair Encoding (BPE)** To address issues with k-mer tokenization, DNABERT2 (Zhou et al., 2024) applies BPE (Sennrich et al., 2016) to DNA. This method iteratively merges the most frequent co-occurring nucleotides into variable-length tokens, enabling the discovery of common sequence motifs while controlling vocabulary growth. This is a popular approach, utilized by other DNA models such as GENA-LM (Fishman et al., 2025) and MistralDNA (Mourad, 2024). However, BPE-tokenized models have shown poor performance on character-level tasks in natural language, such as spelling (Pagnoni et al., 2025). This is a particularly relevant issue in DNA, where letter level single nucleotide variants are critical.

**Learnable tokenization:** Approaches such as VQDNA (Li et al., 2024) and MxDNA (Qiao et al., 2024) learn discrete embeddings or mixture-of-experts assignments for sequence fragments, producing vocabularies tailored to genomic corpora. Although adaptive, these methods introduce additional training and inference overhead while not reducing the input sizes to the transformer, and the learned vocabulary are opaque.

**Single nucleotide:** Despite these innovations, no single tokenization paradigm consistently outperforms others across diverse genomic tasks (Dotan et al., 2024; Lindsey et al., 2025; Patel et al., 2024). Consequently, the canonical nucleotide-level representation is still widely used, for instance in HyenaDNA (Nguyen et al., 2023), Caduceus (Schiff et al., 2024) and the 40B-parameter Evo2 (Brixi et al., 2025). This resolution is essential for fine-grained tasks such as variant effect prediction, which aims to accurately model DNA functional impact Benegas et al. (2025). However, it is computationally inefficient, as genomic sequences are far longer than natural language, and key regulatory elements, such as enhancers, can be over 100kbp from their target genes (Sanyal et al., 2012). Thus, effective sequence compression is critical for scalable DNA modeling.

The approach presented here explores an alternative to tokenization that maintains single-nucleotide granularity, compresses low-information regions, remains interpretable, and allows post-training adaptation. This unique combination of features is unmet by existing methods and yields superior model performance.

## 3 PATCHDNA: BIOLOGICALLY-INFORMED MODELING OF DNA

### 3.1 PATCHING PRELIMINARIES

We follow the patching framework set out by the BLT (Pagnoni et al., 2025). Let $\mathbf{x} = (x_1, x_2, \ldots, x_n)$ be a vector denoting a sequence of $n$ bytes. A patching function is defined as $f_p : \mathbf{x} \mapsto \mathbf{b} \in \{0,1\}^n$, where $b_i = 1$ indicates that position $i$ marks the beginning of a new patch, and $b_i = 0$ otherwise. To ensure existence of at least a single patch we set $b_1 = 1$. This binary sequence $\mathbf{b} = (b_1, b_2, \ldots, b_n)$ partitions the input sequence $\mathbf{x}$ into $m = \sum_{i=1}^{n} b_i$ contiguous subsequences, or *patches*, $\mathbf{p} = (p_1, p_2, \ldots, p_m)$.

We distinguish between tokens and patches in the context of sequence modeling. Tokens are predefined groupings of bytes drawn from a finite vocabulary $\mathcal{V}$, which is determined prior to training. In contrast, patches are variable-length subsequences derived computationally from the input $\mathbf{x}$ by the patching function $f_p$, without relying on a fixed vocabulary.

**Entropy-based patching:** In BLT, patch boundaries are determined dynamically based on predictive uncertainty. Specifically, the patching function relies on the estimated conditional entropy $\hat{H}(x_i \mid x_1, \ldots, x_{i-1})$ computed by a lightweight next-token prediction model. A new patch is initiated when the entropy exceeds a predefined threshold $\theta_H$. Formally, the entropy-based patching function is defined as:

$$f_{\text{entropy}}(x_{i+1}) = \begin{cases} 1 & \text{if } \hat{H}(x_i \mid x_1, \ldots, x_{i-1}) > \theta_H, \\ 0 & \text{otherwise,} \end{cases}$$

The threshold $\theta_H$ controls a tradeoff between granularity and efficiency: lower values yield smaller patches and longer sequences; higher values result in coarser patches and improved efficiency.

**Generalized patching strategy:** We define a flexible class of patching functions $f_p$ where boundaries are determined when the scoring function $g_p$, evaluated over the input sequence, exceeds a

predefined threshold $\theta_p$:

$$f_p(x_{i+1}) = \begin{cases} 1 & \text{if } g_p(x_i) > \theta_p, \\ 0 & \text{otherwise.} \end{cases}$$

Throughout, we use $g_p$ and $\theta_p$ to define the patching strategy.

## 3.2 APPLICATION OF PATCHING FOR DNA MODELING

The novelty of our work lies in demonstrating the unique advantages of patching for DNA language models. While the BLT framework was developed for NLP, its potential extensions to genomics remain unexplored. Unlike BLT, we move beyond entropy-based patching and show that the lack of fixed vocabulary allows more tailored patching functions that can be designed to incorporate domain-specific inductive biases. Leveraging this, we propose biologically informed patching approaches, where we highlight the superiority of conservation-based patching compared to previous methods. We further extend the framework in several ways. First, we introduce *re-patching*, allowing patching strategies to be modified after pretraining, a capability particularly valuable for DNA and potentially other domains. Second, while BLT primarily focuses on generation, we demonstrate that extracting embeddings at single-nucleotide resolution provides unique advantages for genomic analysis. Finally, DNA sequences are much longer than typical NLP inputs. While BLT only considers sequence lengths up to 16k bytes, we process sequences exceeding 100k nucleotides by using larger average patch sizes, yielding far fewer FLOPs than existing DNA models at similar lengths (see Table 18 in Section A.3.4). Achieving equivalent efficiency with tokenization would require 20-mer tokens, leading to an intractable vocabulary of size $4^{20}$. Together, these extensions establish patching as a practical and scalable paradigm for modeling realistic DNA sequences.

## 3.3 CONSERVATION-DRIVEN PATCHING

We apply the generalized patching framework to genomic sequences by treating each byte as one of the four canonical nucleotides (A, C, G, T) or the unknown base N. While entropy-based patching in BLT is motivated by linguistic ambiguity, we hypothesize that in the genomic domain, computational focus should instead align with regions of high evolutionary conservation (Figure 1B).

To implement this, we define the scoring function $g_p$ as the PhyloP conservation score (Pollard et al., 2010; Siepel et al., 2005), a scalar value derived from multi-species alignments (Edgar & Batzoglou, 2006) that quantifies the evolutionary constraint at each nucleotide. In Section 4, we demonstrate that conservation-based patching serves as a strong general-purpose strategy for DNA language models, offering robust performance across diverse downstream tasks.

## 3.4 RE-PATCHING

Genomic tasks often require modeling context or cell-type-specific signals, and the optimal patching strategy may vary by task. As discussed in Section 2, different tokenization schemes can yield varying performance across distinct genomic tasks.

To accommodate this, we introduce *re-patching*, a novel capability to redefine patch boundaries after pretraining. Unlike models constrained by fixed token vocabularies, our approach enables post-hoc modification on the patching function $f_p$, which depends only on the scoring function $g_p$ and threshold $\theta_p$. This makes it straightforward to substitute $g_p$ in inference or fine-tuning time with task- or tissue-specific epigenetic signals, such as chromatin accessibility measured by DNase-seq (Klemm et al., 2019). See Section A.6 for further implementation details. As shown in Section 4.5, this simple adaptation yields substantial gains on cell-type–specific benchmarks, without requiring model retraining from scratch. Importantly, our approach is not constrained by a need for biological information, but can exploit it to guide patching when available and informative. When conservation or other biological signals are absent, the model can readily turn to alternative patching strategies, such as fixed patching, without requiring architectural changes (see Table 12 Section A.3.1).

## 3.5 ARCHITECTURE

The backbone for the work above is the BLT model (Pagnoni et al., 2025), which is an autoregressive model consisting of three main components: a small local encoder, a deep latent global transformer, and a small local decoder.

**Local encoder**: This is a shallow transformer that computes patch-level representations from a single-nucleotide input sequence $\mathbf{x}$, using patch boundaries provided by the patching function $f_p$. It alternates between sliding window self-attention layers (operating over the nucleotide sequence) and cross-attention layers, following the Perceiver architecture (Jaegle et al., 2021). Patch representations are queries, which attend only to the nucleotides (keys) within their respective patch.

**Latent global transformer**: This is a standard transformer (Vaswani et al., 2017), using rotary positional encodings (Su et al., 2024), operating on the patch embeddings produced by the local encoder. It models long-range interactions across the full sequence using global attention. Since the patch sequence $\mathbf{p}$ is much shorter than the input sequence $\mathbf{x}$, this module can be made significantly deeper, allowing the bulk of the model's capacity to focus on global reasoning without incurring prohibitive computational cost.

**Local decoder**: This lightweight transformer updates the nucleotide-level representations from the local encoder to incorporate the patch embedding output from the global transformer. Like the local encoder, it alternates between sliding window self-attention and cross-attention layers. In this case, the single-nucleotide embeddings serve as queries, while the patch embeddings act as keys and values. A language modeling head is applied to the final nucleotide embeddings to produce logits for next-nucleotide prediction during autoregressive pretraining.

### 3.5.1 PRETRAINING AND DOWNSTREAM USAGE

We pretrain `PatchDNA` on the human reference genome using a next-nucleotide prediction objective, following the same training and validation splits as Caduceus (Schiff et al., 2024) and HyenaDNA (Nguyen et al., 2023), as originally defined by (Kelley, 2020). During pretraining, we set the patching threshold $\theta_p$ to the 95th percentile of the scoring function $g_p$ (based on PhyloP conservation scores, or entropy), resulting in an average patch size of approximately 20 nucleotides. See Section A.7 for results using other conservation scoring and sensitivity analysis at other thresholds. This enables efficient training with input contexts up to 131,000 base pairs. To our knowledge, this is the first transformer-based architecture in DNA language modeling capable of efficiently handling such long sequences at scale. We pretrain two main models: `PatchDNA`, a 19.2M parameter model with a 16 kbp context window, and `PatchDNA-7M`, a 7.7M parameter model with a 131 kbp context window. The latter is designed to enable fairer comparisons with other long-range sequence models, such as Caduceus (7.7M) and HyenaDNA (6.6M). We set a maximum patch size to prevent over-compression of the DNA sequence in non-conserved regions. Full hyperparameter and training details are provided in Section A.2.

While the original BLT paper focused on generation tasks in natural language processing, we show that when pretrained on genomic sequences, the decoder's nucleotide-level embeddings yield meaningful representations for a wide range of downstream tasks. These embeddings retain single-nucleotide resolution, making them particularly well suited for fine-grained genomic prediction problems. For all downstream applications, we extract the penultimate layer of the decoder as a nucleotide-level embedding representation.

## 4 EXPERIMENTS

We compare against a range of strong baselines (see Section A.1), including small models such as HyenaDNA (Nguyen et al., 2023) and Caduceus (Schiff et al., 2024) both with around 7 million parameters, as well as large-scale DNA models ranging from 110 million to 500 million parameters, including GENA-LM (Fishman et al., 2025), DNABERT2 (Zhou et al., 2024), MistralDNA (Mourad, 2024) and the Nucleotide Transformer variants (Dalla-Torre et al., 2025). In Section A.4, we present extensive ablations where we compare to PatchDNA-Entropy and PatchDNA-FixedPS20 models pretrained and evaluated with entropy and fixed patching.

## 4.1 NUCLEOTIDE TRANSFORMER BENCHMARK

The NT benchmark spans 18 supervised classification tasks (300–1000 bp sequences) across three categories: regulatory element detection, splice site prediction, and chromatin profile annotation. Each task is framed as a supervised classification problem, and all models are evaluated using a standardized protocol repeated across five random seeds. Specifically, a frozen pretrained model encodes each DNA sequence into a latent embedding space, and a linear probe is trained on top of these fixed representations, similar to (Marin et al., 2024). This setup enables a controlled comparison of representational quality irrespective of the underlying architecture.

Figure 2 shows mean Matthews Correlation Coefficient (MCC) per category. `PatchDNA` achieves the highest average MCC in regulatory elements and splicing tasks, and remains competitive on chromatin profile classification, matching larger-scale models such as `NT-MS-500M`. Detailed results for all 18 tasks are given in Section A.3.1. We further show that `PatchDNA` outperforms strong baselines under finetuning (Section A.3.1), in addition to the probing results reported here.

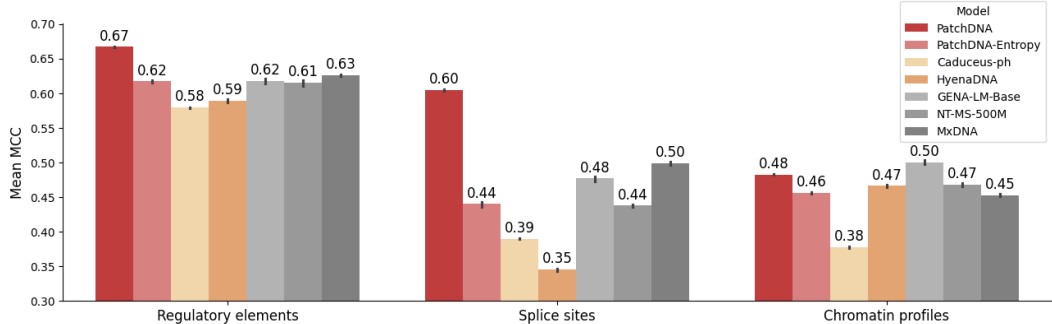

Figure 2: Mean MCC across task categories on the NT benchmark. Models are grouped by size: orange shades indicate small models, and grey shades represent large models. Error bars denote one standard deviation across five seeds.

## 4.2 DART-EVAL BENCHMARK

Next, we evaluate our model on DART-Eval (Patel et al., 2024), a benchmark covering five regulatory genomics tasks. These include distinguishing regulatory sequences from matched controls (Task 1), detecting transcription factor (TF) motifs (Task 2), identifying cell-type-specific signatures (Task 3), predicting regulatory activity levels (Task 4), and variant effect prediction (Task 5). The benchmark combines both classification and regression tasks, with settings that test zero-shot capabilities and supervised probing.

We use the official DART-Eval implementation and adopt the *zero-shot* configuration wherever it is available, specifically for Tasks 1, 2 and 5. These tasks are evaluated directly using model likelihoods or embeddings, without any additional training. For Tasks 3 and 4, which lack zero-shot variants, we follow the standard protocol and train probes on top of frozen embeddings. For Tasks 4 and 5, where multiple sub tasks exist, we report the mean across the tasks. Detailed results for sub tasks can be found in the A.3.2.

For competing models, we report the values given in the original benchmark, which have been performed on one seed. As shown in Table 1, our model achieves the best overall performance on DART-Eval, with the best mean rank (2) across all five tasks. `PatchDNA-Entropy` is second best on the benchmark, highlighting the benefit of patching and the BLT architecture independently of conservation-based patching. While other models show strength on individual tasks, such as `NT-MS-500M` on Task 5 or `HyenaDNA` on Task 3, they do not generalize as broadly.

## 4.3 BEND

The BEND benchmark (Marin et al., 2024) combines sequence-level classification tasks, such as chromatin accessibility, histone modification, and CpG methylation, with nucleotide-level classifi-

Table 1: Performance on the DART-Eval benchmark. Raw task metrics are shown, taking the mean across sub tasks for Task 4 and Task 5. Overall mean rank across all tasks is computed in the final column.

| Model | Task 1 | Task 2 | Task 3 | Task 4 | Task 5 | Mean rank |
|-------|--------|--------|--------|--------|--------|-----------|
| | Accuracy | Accuracy | Accuracy | Spearman $R$ | AUROC | |
| PatchDNA | 0.966 | **0.725** | 0.457 | 0.440 | 0.555 | **2.0** |
| PatchDNA-Entropy | 0.965 | 0.650 | 0.465 | 0.400 | 0.523 | 3.0 |
| HyenaDNA | 0.891 | 0.645 | **0.587** | 0.384 | 0.515 | 3.8 |
| GENA-LM-Large | 0.947 | 0.620 | 0.383 | **0.472** | 0.505 | 4.2 |
| NT-MS-500M | 0.745 | 0.565 | 0.420 | 0.422 | **0.566** | 4.8 |
| Caduceus-ps | **0.971** | 0.570 | 0.281 | 0.297 | 0.514 | 5.8 |
| DNABERT2 | 0.876 | 0.590 | 0.371 | 0.419 | 0.493 | 6.0 |
| MistralDNA | 0.863 | 0.625 | 0.329 | 0.363 | 0.498 | 6.4 |

cation tasks like gene finding. Gene finding requires multi-class annotation of each base over sequences up to 14 kbp, making it a fine-grained task that depends on both local context and long-range dependencies. Our patching strategy, combined with cross-attention, enables precise nucleotide representations while flexibly aggregating context.

We follow the original BEND evaluation protocol, by training a probe on top of frozen embeddings. For competing models, we report the values given in the original benchmark, which have been performed on one seed. As shown in Table 2, `PatchDNA` achieves consistently strong performance, outperforming other models in 3 out of 4 tasks. On the gene finding task, it outperforms larger models such as `GENA-LM-Large` and `DNABERT2` ranking second only to `NT-MS-500M`; a model with 25 fold greater capacity (500M vs 19.2M parameters), and pre-trained on a substantially larger, multi-species dataset. Results on the remaining tasks in BEND are reported in Section A.3.3.

Table 2: Performance across BEND short and long range tasks. Gene finding is reported with MCC, while other tasks are reported with AUROC.

| Model | Gene finding | Chromatin accessibility | Histone modification | CpG Methylation |
|-------|--------------|-------------------------|----------------------|-----------------|
| | MCC | AUROC | AUROC | AUROC |
| PatchDNA | 0.58 | **0.84** | **0.79** | **0.92** |
| PatchDNA-Entropy | 0.37 | 0.83 | 0.78 | 0.90 |
| NT-MS-500M | **0.64** | 0.80 | 0.76 | 0.91 |
| GENA-LM-Large | 0.52 | 0.76 | 0.78 | 0.91 |
| Caduceus-ph | 0.44 | 0.80 | **0.79** | 0.90 |
| HyenaDNA | 0.35 | **0.84** | 0.76 | 0.91 |
| DNABERT-2 | 0.43 | 0.81 | 0.78 | 0.90 |

## 4.4 CAGE PREDICTION BENCHMARK

To evaluate performance on long DNA sequences, we benchmark `PatchDNA` on CAGE prediction (Avsec et al., 2021). CAGE (Cap Analysis of Gene Expression) quantifies gene expression and identifies transcription start sites. The prediction task involves regressing expression values across bins in a 114,688 bp input sequence, leveraging distal regulatory elements that may lie kilobases away from the target gene.

We follow the setup from (Trop et al., 2025), using 50 CAGE tracks and the full 114 kbp context window. We only compare to other DNA language models that can handle such long sequences in one forward pass. For fair comparison, we use the `PatchDNA-7M` model to match the parameter budget of `HyenaDNA` and `Caduceus`. All models are fine-tuned for one epoch using an MLP head and evaluated using Pearson correlation at the gene, cell, and full-track levels, following the metrics introduced in Enformer (Avsec et al., 2021). We give detailed explanations of these metrics in Supplementary A.3.4.

As shown in Table 3, `PatchDNA-7M` outperforms all baselines across evaluation metrics, achieving the highest gene- and cell-level Pearson correlations. To further boost performance, we introduce a

variant that adjusts the patching strategy during fine-tuning by leveraging cCRE annotations (Moore et al., 2020) to focus attention on known regulatory regions. This modification, which is applied only at fine-tuning time, and can only be done with `PatchDNA`, leads to additional gains. This demonstrates that our framework can flexibly incorporate biological priors without requiring model retraining or changes to the underlying architecture. `PatchDNA` also offers practical efficiency advantages, finetuning more than $3\times$ faster than `HyenaDNA` (see Table 18, Section A.3.4), highlighting the benefit of moving beyond single-nucleotide tokenization. For results on other long range tasks, see Section A.3.6. `PatchDNA` outperforms other long sequence models on 6 out of 7 tasks in the Genomics Long Range Benchmark (Trop et al., 2025).

Table 3: Performance on the CAGE prediction task. We report mean Pearson correlation across genes, cells, and full sequence bins. Error bars denote one standard deviation across five seeds.

| Model | Gene Pearson | Cell Pearson | Full Pearson |
|---|---|---|---|
| PatchDNA-7M | $\underline{0.369} \pm 0.001$ | $\underline{0.771} \pm 0.002$ | $\mathbf{0.471} \pm 0.002$ |
| PatchDNA-7M + cCRE-aware re-patching | $\mathbf{0.373} \pm 0.001$ | $\mathbf{0.792} \pm 0.002$ | $0.408 \pm 0.004$ |
| HyenaDNA | $0.362 \pm 0.001$ | $0.745 \pm 0.002$ | $0.290 \pm 0.004$ |
| Caduceus-ph | $0.362 \pm 0.001$ | $0.750 \pm 0.002$ | $0.309 \pm 0.003$ |
| Caduceus-ps | $0.365 \pm 0.001$ | $0.766 \pm 0.001$ | $\underline{0.420} \pm 0.006$ |

## 4.5 Cell type specific re-patching

Because the DNA sequence is invariant across cell types, sequence-only models often struggle with context-specific tasks such as predicting cell-type-specific expression (Patel et al., 2024). We show that our model can be adapted to such tasks with minimal modification and without changing the architecture or retraining from scratch. Using the setup in Section 4.4, we evaluate performance on CAGE prediction across three cell types: K562, hepatocytes, and neurons. For each task, we predict expression for a single CAGE track corresponding to the target cell type.

Cell-type-specific epigenetic inputs like DNase-seq data can help provide cellular context by highlighting regulatory regions of the genome that are accessible and potentially active in transcription (Carter & Zhao, 2021). While previous methods like EPInformer (Lin et al., 2024) and Seq2Exp (Su et al., 2025) rely on custom architectures that fuse sequence with epigenetic inputs, we instead only re-patch the DNA using DNase-seq signal from the target cell type. This only alters the patches, preserving the underlying model architecture while focusing computation on regulatory regions inferred from chromatin accessibility.

As shown in Table 4, `PatchDNA` outperforms all competing baselines on cell type-specific CAGE prediction. Given that Caduceus-ps outperforms Caduceus-ph in Section 4.4, we only compare to Caduceus-ps in this task. Incorporating DNase-aware patching further improves performance across all three cell types, demonstrating that context-specific patching is highly informative for modeling regulatory activity. Table 5 shows that these gains are maximized when the DNase-seq signal used for patching matches the target tissue. In contrast, mismatched signals lead to consistently lower performance, highlighting the importance of aligning the patching strategy with the underlying cellular context. Notably, these improvements are achieved without altering the model architecture or retraining from scratch.

Table 4: Performance on cell type-specific CAGE prediction, reported as Pearson correlation across cells. Error bars denote one standard deviation across five seeds.

| Model | K562 | Hepatocyte | Neuron |
|---|---|---|---|
| PatchDNA-7M | $\underline{0.754} \pm 0.003$ | $\underline{0.717} \pm 0.002$ | $\underline{0.799} \pm 0.001$ |
| PatchDNA-7M + DNase-aware re-patching | $\mathbf{0.828} \pm 0.001$ | $\mathbf{0.727} \pm 0.001$ | $\mathbf{0.831} \pm 0.001$ |
| HyenaDNA | $0.703 \pm 0.012$ | $0.667 \pm 0.006$ | $0.763 \pm 0.003$ |
| Caduceus-ps | $0.732 \pm 0.006$ | $0.705 \pm 0.001$ | $0.798 \pm 0.002$ |

Table 5: Performance on DNase-aware cell type-specific CAGE prediction, reported as Pearson correlation across cells. Maximum performance is achieved when patching is guided by DNase-seq signal from the corresponding tissue (the diagonal), and applied during fine-tuning. Error bars denote one standard deviation across five seeds.

| Model | K562 | Hepatocyte | Neuron |
|---|---|---|---|
| PatchDNA-7M DNase-aware (K562) | **0.828** $\pm$ 0.001 | 0.713 $\pm$ 0.001 | 0.807 $\pm$ 0.002 |
| PatchDNA-7M DNase-aware (Hepatocyte) | 0.775 $\pm$ 0.002 | **0.727** $\pm$0.001 | 0.822 $\pm$ 0.001 |
| PatchDNA-7M DNase-aware (Neuron) | 0.770 $\pm$ 0.001 | 0.707 $\pm$ 0.001 | **0.831** $\pm$0.001 |

## 5 DISCUSSION

We introduce `PatchDNA`, a novel DNA language modeling framework that replaces fixed tokenization with a dynamic patching mechanism, which improves efficiency, enables models to focus adaptively on functionally relevant genomic regions, and provides flexibility through re-patching. By introducing conservation-driven and context-aware patching strategies, `PatchDNA` allocates model capacity to the most informative regions of the genome, without relying on fixed vocabularies. Beyond pretraining, `PatchDNA` introduces *re-patching*: the ability to redefine patch boundaries post hoc. This property allows our model to use tissue-specific or task-specific signals to adapt to downstream tasks, such as cell-type–specific expression prediction, without retraining from scratch. Furthermore, our framework supports re-patching with alternative strategies when biological signals are unavailable, demonstrating that it is not dependent on such inputs but can flexibly exploit them whenever they are present and informative.

Through extensive benchmarking, we demonstrate that `PatchDNA` consistently outperforms or matches state-of-the-art models across regulatory element prediction, splicing, and gene expression tasks, while training significantly faster. Complementary ablations further highlight the effectiveness of our approach: conservation-guided patching outperforms entropy-based and fixed-size baselines, as well as using PhyloP conservation scores directly (see Section A.4). Notably, while raw conservation scores are only weakly correlated with task labels in most benchmarks, conservation-based patching still yields substantial gains, demonstrating that PatchDNA extracts richer, functionally grounded representations than conservation scores alone.

### 5.1 LIMITATIONS AND FUTURE WORK

While `PatchDNA` offers a versatile framework for DNA modeling, several limitations remain. First, the architecture we used is autoregressive and decoder-only, which limits its ability to fully capture the bidirectional context that is often critical in genomics (Schiff et al., 2024). Extending the framework to support bidirectional encoding could further improve performance on context-rich tasks (Schiff et al., 2024; Schmidinger et al., 2025). Second, we currently pretrain only on the human reference genome. Incorporating multi-species data (see Section A.5) or genetic variation from population-scale datasets could expand the model's applicability and improve generalization to unseen genomic contexts (Brixi et al., 2025; Dalla-Torre et al., 2025). Another open direction is the incorporation of reverse-complement (RC) equivariance, which is a desirable inductive bias in DNA modeling (Mallet & Vert, 2021).

While we demonstrate re-patching on selected tasks, future work should evaluate the generality of this mechanism across a broader range of biological applications, including regulatory activity prediction and variant effect interpretation (Avsec et al., 2021; Linder et al., 2025). `PatchDNA` provides a modular foundation to explore these extensions with minimal architectural changes. Furthermore, assessing the scaling-law behavior of these models and comparing their performance to existing approaches will be an important avenue for future work (Nguyen et al., 2024).

We hope that our general framework will serve as a foundation for future work, inspiring the development of new patching strategies and advancing the broader field of DNA language models through task- and biology-aware modeling in contrast to the current emphasis on scaling laws Brixi et al. (2025).

## 6 REPRODUCIBILITY STATEMENT

We provide detailed hyperparameters and setup for pretraining the PatchDNA models in Section A.2. For downstream tasks, we provide the methodology we use in A.3, where we default to established practices in the literature where available. All datasets that we use are publicly available and links are given in each section where we use external datasets (Section A.2, Section A.3). All baselines that we use from literature can be downloaded from publicly available sources, with links given in Section A.1.

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

# A  APPENDIX

## A.1  DETAILS OF PRETRAINED BASELINE MODELS

Table 6: Overview of pretrained DNA language models used in this study. We list HuggingFace IDs, number of parameters, and species coverage.

| Model | HuggingFace ID | Parameters | Species |
|---|---|---|---|
| HyenaDNA | LongSafari/hyenadna-large-1m-seqlen-hf | 6.6M | Human |
| Caduceus-ps | kuleshov-group/caduceus-ps_seqlen-131k_d_model-256_n_layer-16 | 7.7M | Human |
| Caduceus-ph | kuleshov-group/caduceus-ph_seqlen-131k_d_model-256_n_layer-16 | 7.7M | Human |
| DNABERT2 | zhihan1996/DNABERT-2-117M | 117M | Multispecies |
| GENA-LM-Base | AIRI-Institute/gena-lm-bert-base-t2t | 110M | Human |
| GENA-LM-Large | AIRI-Institute/gena-lm-bert-large-t2t | 336M | Multi-species |
| MistralDNA | RaphaelMourad/Mistral-DNA-v1-1.6B-hg38 | 1.6B | Human |
| NT-MS-500M | InstaDeepAI/nucleotide-transformer-v2-500m-multi-species | 500M | Multi-species |
| NT-MS-100M | InstaDeepAI/nucleotide-transformer-v2-100m-multi-species | 100M | Multi-species |
| MxDNA | github.com/qiaoqiaoLF/MxDNA/tree/full-model | 100M | Human |

## A.2  PRETRAINING DETAILS

### ARCHITECTURE HYPERPARAMETERS

Table 7: Architecture hyperparameters for PatchDNA and PatchDNA-7M. The patching threshold is the 95% quantile of all PhyloP scores

| Hyperparameter | PatchDNA | PatchDNA-7M |
|---|---|---|
| Num Local Encoder Layers | 4 | 2 |
| Num Local Decoder Layers | 4 | 2 |
| Num Global Transformer Layers | 8 | 3 |
| Embedding Dimension | 256 | 256 |
| Context Length | 16,000 | 131,072 |
| Max Patch Length | 128 | 1,024 |
| Number of Global Transformer Heads | 8 | 4 |
| Number of Local Encoder Heads | 8 | 4 |
| Number of Local Decoder Heads | 8 | 4 |
| PhyloP Patching Threshold | 1.5 | 1.5 |
| Num parameters | 19.2M | 7.7M |

### TRAINING HYPERPARAMETERS

We use the same optimizer, learning rate, weight decay, and gradient clipping as Pagnoni et al. (2025).

Table 8: Training hyperparameters for PatchDNA and PatchDNA-7M.

| Hyperparameter | PatchDNA | PatchDNA-7M |
|---|---|---|
| Learning Rate | 0.0004 | 0.0004 |
| Training Steps | 100,000 | 100,000 |
| Weight Decay | 0.1 | 0.1 |
| Optimizer | AdamW | AdamW |
| Batch Size | 64 | 8 |
| Gradient Clipping | 1.0 | 1.0 |
| Training Time (4×A100 80GB) | ∼18 hours | ∼10 hours |

PATCHING ABLATION CONFIGURATIONS

- **PatchDNA Entropy:** Uses identical hyperparameters to PatchDNA, except it employs a small entropy model for patching with a threshold of 1.37 (which is 95% quantile of all scores from the entropy model across the genome). Hyperparameter details for the entropy model are in Table 9.
- **PatchDNA Fixed Patch Size 20:** Shares the same hyperparameters as PatchDNA, but uses a fixed patch size of 20. i.e., every 20 nucleotides are in one patch. We use this because a patching threshold of the 95% quantile of all scores gives an average patch size of approximately 20.

Table 9: Hyperparameters for the entropy model used in PatchDNA Entropy.

| Hyperparameter | Value |
|---|---|
| Number of Layers | 8 |
| Embedding Dimension | 256 |
| Context Length | 8,192 |
| Sliding window | 512 |
| Number of Heads | 8 |
| Batch size | 256 |
| Learning Rate | 0.0004 |
| Training Steps | 100,000 |
| Weight Decay | 0.1 |
| Optimizer | AdamW |
| Gradient Clipping | 1.0 |
| Num parameters | 6.8M |

DATA

We use the same train and validation splits as HyenaDNA (Nguyen et al., 2023) and Caduceus (Schiff et al., 2024), which originate from Kelley (2020), available at https://console.cloud.google.com/storage/browser/basenji_barnyard/data

We use the PhyloP scores (Siepel et al., 2005; Pollard et al., 2010) downloaded from https://hgdownload.cse.ucsc.edu/goldenpath/hg38/phyloP100way/

CODE

Code will be available at https://github.com/RelationRx/patchDNA

## A.3 BENCHMARK TASKS DETAILS

### A.3.1 NUCLEOTIDE TRANSFORMER BENCHMARK

We evaluated model performance on the Nucleotide Transformer (NT) benchmark, a diverse collection of 18 classification tasks designed to assess the biological utility of pretrained DNA language models. The benchmark was accessed via the HuggingFace Hub[1], and includes pre-defined train and test splits for each task. For each task, we further partitioned the provided training set into 90% training and 10% validation splits. All experiments were repeated across five random seeds, with each seed generating a new train/validation split to evaluate consistency and robustness.

To ensure fair and consistent evaluation across models, we adopted a linear probing protocol. Specifically, each pretrained model was frozen and used to encode input DNA sequences into latent embeddings, over which a linear classifier was trained. The input representation dimensionality varied across models: `PatchDNA`, `Caduceus-ph` and `HyenaDNA` produced 256-dimensional embeddings, while `GENA-LM-Base` and `NT-MS-500M` yielded 768 and 1024-dimensional embeddings, respectively.

---

[1]https://huggingface.co/datasets/InstaDeepAI/nucleotide_transformer_downstream_tasks_revised

All models were evaluated under identical training conditions: a batch size of 64, a total of 50 training epochs, and optimization using AdamW with a learning rate of $5e - 4$ and weight decay of $0.01$. For each model and seed, we report performance on the official test set using Matthews Correlation Coefficient (MCC), averaged across all runs. Full per-task results with standard deviations are presented in Supplementary Table 10.

Table 10: Detailed performance across all 18 tasks in the Nucleotide Transformer Benchmark.

| Dataset / Model | promoter_all | promoter_no_tata | promoter_tata | enhancers | enhancers_types | splice_sites_acceptors |
|---|---|---|---|---|---|---|
| PatchDNA | **0.779 ± 0.007** | **0.786 ± 0.003** | **0.853 ± 0.009** | 0.475 ± 0.004 | 0.441 ± 0.005 | **0.669 ± 0.006** |
| PatchDNA Entropy | 0.719 ± 0.007 | 0.743 ± 0.003 | 0.749 ± 0.04 | 0.454 ± 0.01 | 0.421 ± 0.008 | 0.497 ± 0.005 |
| Caduceus-ph | 0.679 ± 0.001 | 0.727 ± 0.002 | 0.67 ± 0.0 | 0.429 ± 0.002 | 0.39 ± 0.002 | 0.448 ± 0.002 |
| HyenaDNA | 0.712 ± 0.002 | 0.729 ± 0.001 | 0.71 ± 0.009 | 0.414 ± 0.005 | 0.38 ± 0.004 | 0.391 ± 0.009 |
| GENA-LM-Base | 0.7 ± 0.008 | 0.741 ± 0.013 | 0.707 ± 0.02 | **0.488 ± 0.01** | **0.452 ± 0.008** | 0.54 ± 0.007 |
| NT-MS-500M | 0.718 ± 0.003 | 0.741 ± 0.004 | 0.685 ± 0.032 | 0.485 ± 0.003 | 0.445 ± 0.003 | 0.468 ± 0.005 |
| MxDNA | 0.729 ± 0.006 | 0.757 ± 0.007 | 0.759 ± 0.011 | 0.458 ± 0.004 | 0.428 ± 0.004 | 0.57 ± 0.004 |

| Dataset / Model | splice_sites_all | splice_sites_donors | H2AFZ | H3K27ac | H3K27me3 | H3K36me3 |
|---|---|---|---|---|---|---|
| PatchDNA | **0.454 ± 0.018** | **0.692 ± 0.014** | 0.396 ± 0.005 | 0.41 ± 0.022 | 0.557 ± 0.004 | 0.542 ± 0.004 |
| PatchDNA Entropy | 0.311 ± 0.011 | 0.512 ± 0.007 | 0.401 ± 0.007 | 0.352 ± 0.008 | 0.529 ± 0.004 | 0.498 ± 0.006 |
| Caduceus-ph | 0.267 ± 0.002 | 0.455 ± 0.003 | 0.337 ± 0.002 | 0.28 ± 0.003 | 0.476 ± 0.003 | 0.354 ± 0.003 |
| HyenaDNA | 0.258 ± 0.008 | 0.387 ± 0.005 | **0.444 ± 0.004** | 0.375 ± 0.003 | 0.507 ± 0.002 | 0.498 ± 0.001 |
| GENA-LM-Base | 0.312 ± 0.005 | 0.578 ± 0.007 | 0.403 ± 0.012 | **0.449 ± 0.01** | **0.565 ± 0.013** | **0.553 ± 0.006** |
| NT-MS-500M | 0.336 ± 0.005 | 0.509 ± 0.004 | 0.392 ± 0.005 | 0.398 ± 0.004 | 0.536 ± 0.004 | 0.496 ± 0.006 |
| MxDNA | 0.34 ± 0.006 | 0.585 ± 0.006 | 0.366 ± 0.004 | 0.363 ± 0.004 | 0.532 ± 0.003 | 0.474 ± 0.006 |

| Dataset / Model | H3K4me1 | H3K4me2 | H3K4me3 | H3K9ac | H3K9me3 | H4K20me1 |
|---|---|---|---|---|---|---|
| PatchDNA | 0.406 ± 0.009 | 0.459 ± 0.004 | 0.614 ± 0.006 | 0.47 ± 0.011 | 0.393 ± 0.012 | 0.576 ± 0.008 |
| PatchDNA Entropy | 0.381 ± 0.009 | 0.457 ± 0.013 | 0.583 ± 0.006 | 0.458 ± 0.023 | 0.346 ± 0.009 | 0.554 ± 0.005 |
| Caduceus-ph | 0.333 ± 0.001 | 0.403 ± 0.005 | 0.489 ± 0.002 | 0.379 ± 0.005 | 0.214 ± 0.007 | 0.513 ± 0.004 |
| HyenaDNA | 0.387 ± 0.004 | **0.493 ± 0.006** | **0.627 ± 0.005** | 0.485 ± 0.004 | 0.291 ± 0.013 | 0.554 ± 0.004 |
| GENA-LM-Base | **0.42 ± 0.01** | 0.486 ± 0.006 | 0.624 ± 0.011 | 0.5 ± 0.005 | **0.4 ± 0.013** | **0.604 ± 0.01** |
| NT-MS-500M | 0.391 ± 0.009 | 0.47 ± 0.005 | 0.622 ± 0.007 | **0.514 ± 0.005** | 0.304 ± 0.016 | 0.561 ± 0.001 |
| MxDNA | 0.387 ± 0.004 | 0.458 ± 0.003 | 0.568 ± 0.0 | 0.463 ± 0.013 | 0.36 ± 0.016 | 0.559 ± 0.003 |

FINETUNING RESULTS

Using the same finetuning protocol and hyperparameters as (Qiao et al., 2024), we fully finetune all models across 3 seeds for a maximum of 20 epochs. The results in Table 11 show that PatchDNA performs strongly across the benchmark, outperforming all other models in 11 out of 18 tasks.

Table 11: Comparison of finetuning results on NT benchmark.

| Dataset Model | promoter_all | promoter_no_tata | promoter_tata | enhancers | enhancers_types | splice_sites_acceptors |
|---|---|---|---|---|---|---|
| PatchDNA | **0.791 ± 0.009** | **0.788 ± 0.005** | **0.84 ± 0.019** | **0.528 ± 0.009** | **0.496 ± 0.008** | 0.754 ± 0.04 |
| PatchDNA Entropy | 0.725 ± 0.008 | 0.73 ± 0.007 | 0.785 ± 0.016 | 0.523 ± 0.001 | 0.488 ± 0.008 | 0.868 ± 0.015 |
| Caduceus-ps | 0.742 ± 0.01 | 0.764 ± 0.013 | 0.761 ± 0.028 | 0.51 ± 0.017 | 0.471 ± 0.006 | 0.765 ± 0.006 |
| HyenaDNA | 0.693 ± 0.007 | 0.724 ± 0.004 | 0.831 ± 0.057 | 0.479 ± 0.005 | 0.45 ± 0.003 | 0.82 ± 0.015 |
| GENA-LM-Base | 0.738 ± 0.007 | 0.736 ± 0.025 | 0.689 ± 0.038 | 0.483 ± 0.023 | 0.467 ± 0.012 | 0.76 ± 0.005 |
| NT-MS-100M | 0.737 ± 0.019 | 0.756 ± 0.003 | 0.818 ± 0.052 | 0.513 ± 0.001 | 0.478 ± 0.002 | **0.952 ± 0.002** |
| MxDNA | 0.734 ± 0.013 | 0.755 ± 0.01 | 0.831 ± 0.038 | 0.519 ± 0.014 | 0.48 ± 0.01 | 0.812 ± 0.032 |
| PhyloP | 0.405 ± 0.002 | 0.393 ± 0.006 | 0.469 ± 0.006 | 0.181 ± 0.007 | 0.167 ± 0.002 | 0.543 ± 0.001 |

| Dataset Model | splice_sites_all | splice_sites_donors | H2AFZ | H3K27ac | H3K27me3 | H3K36me3 |
|---|---|---|---|---|---|---|
| PatchDNA | 0.76 ± 0.019 | 0.706 ± 0.026 | **0.523 ± 0.01** | 0.486 ± 0.015 | **0.607 ± 0.008** | **0.621 ± 0.007** |
| PatchDNA Entropy | 0.884 ± 0.013 | 0.654 ± 0.016 | 0.521 ± 0.009 | 0.484 ± 0.035 | 0.595 ± 0.004 | 0.584 ± 0.02 |
| Caduceus-ps | 0.796 ± 0.021 | 0.771 ± 0.013 | 0.507 ± 0.007 | 0.475 ± 0.021 | 0.591 ± 0.009 | 0.607 ± 0.008 |
| HyenaDNA | 0.849 ± 0.006 | 0.84 ± 0.029 | 0.481 ± 0.005 | 0.44 ± 0.003 | 0.554 ± 0.014 | 0.549 ± 0.002 |
| GENA-LM-Base | 0.764 ± 0.013 | 0.781 ± 0.004 | 0.466 ± 0.035 | 0.495 ± 0.01 | 0.588 ± 0.004 | 0.602 ± 0.021 |
| NT-MS-100M | **0.966 ± 0.0** | **0.962 ± 0.003** | 0.501 ± 0.009 | **0.496 ± 0.009** | 0.599 ± 0.009 | 0.617 ± 0.004 |
| MxDNA | 0.86 ± 0.007 | 0.931 ± 0.021 | 0.512 ± 0.003 | 0.489 ± 0.031 | 0.599 ± 0.015 | 0.618 ± 0.002 |
| PhyloP | 0.283 ± 0.004 | 0.547 ± 0.001 | -0.017 ± 0.062 | 0.105 ± 0.028 | 0.233 ± 0.032 | 0.304 ± 0.003 |

| Dataset Model | H3K4me1 | H3K4me2 | H3K4me3 | H3K9ac | H3K9me3 | H4K20me1 |
|---|---|---|---|---|---|---|
| PatchDNA | 0.48 ± 0.003 | **0.573 ± 0.004** | **0.634 ± 0.005** | **0.569 ± 0.01** | 0.47 ± 0.017 | 0.637 ± 0.007 |
| PatchDNA Entropy | 0.472 ± 0.011 | 0.568 ± 0.021 | 0.589 ± 0.01 | 0.546 ± 0.009 | **0.473 ± 0.019** | 0.626 ± 0.027 |
| Caduceus-ps | 0.471 ± 0.014 | 0.565 ± 0.008 | 0.617 ± 0.009 | 0.526 ± 0.009 | 0.435 ± 0.015 | 0.639 ± 0.009 |
| HyenaDNA | 0.438 ± 0.007 | 0.523 ± 0.025 | 0.618 ± 0.007 | 0.497 ± 0.014 | 0.371 ± 0.026 | 0.617 ± 0.008 |
| GENA-LM-Base | 0.465 ± 0.014 | 0.538 ± 0.027 | 0.61 ± 0.055 | 0.525 ± 0.007 | 0.44 ± 0.009 | 0.644 ± 0.011 |
| NT-MS-100M | 0.487 ± 0.01 | 0.551 ± 0.005 | 0.624 ± 0.003 | 0.531 ± 0.002 | 0.469 ± 0.006 | **0.646 ± 0.01** |
| MxDNA | **0.497 ± 0.001** | 0.563 ± 0.012 | 0.627 ± 0.017 | 0.534 ± 0.015 | 0.467 ± 0.023 | **0.646 ± 0.007** |
| PhyloP | 0.006 ± 0.041 | -0.02 ± 0.058 | 0.009 ± 0.084 | 0.026 ± 0.041 | 0.072 ± 0.035 | 0.059 ± 0.103 |

On splice sites, (Lindsey et al., 2025) show that a model trained with single nucleotide tokenization significantly outperforms an equivalent BPE based model. They also postulate that consistent token size facilitates the model's learning of specific distances for these tasks. Inspired by this, we re-patch the model after pretraining, using single nucleotide patching with the same base model. Table 12 shows the expected improvement demonstrated by prior work. Although single-nucleotide patching introduces additional computational overhead during fine-tuning, re-patching avoids the costly requirement of pretraining from scratch at this resolution.

Table 12: Comparison of PatchDNA vs PatchDNA with single nucleotide re-patching on splice site tasks.

| | **PatchDNA** | **PatchDNA re-patch size 1** |
|---|---|---|
| **Pre-training patching** | PhyloP | PhyloP |
| **Finetuning Patching** | PhyloP | Single Nucleotide |
| splice_sites_acceptors | 0.754 ± 0.040 | **0.946 ± 0.002** |
| splice_sites_all | 0.760 ± 0.019 | **0.953 ± 0.006** |
| splice_sites_donors | 0.706 ± 0.026 | **0.948 ± 0.002** |

### A.3.2 DART-EVAL

We evaluated our model's performance by adding it to each task using the original evaluation code provided by the authors at https://github.com/kundajelab/DART-Eval. To ensure consistency, we maintained the original experimental setup and report the published results for all other baseline models directly from the original paper (Patel et al., 2024). We use 1 A100 80GB GPU for each task.

For Task 1 and Task 2, we use the zero-shot likelihoods formulation, while for Task 5, we apply the zero-shot embeddings approach. When both likelihoods and embeddings could be used, we choose

between them based on the relative performance of models across tasks. For example, in Task 2, embeddings from all DNA models perform significantly worse than likelihoods, making the latter the preferred choice. For Task 2, we report median accuracy.

For Task 3 and Task 4, where no zero-shot formulation exists, a lightweight probe is trained on top of frozen model embeddings.

For Task 2, no conservation scores are available, so we re-patch to single nucleotide scores at inference time, showing the flexibility of our modeling approach when conservation scores are unavailable.

VISUALIZATION OF OVERALL PERFORMANCE

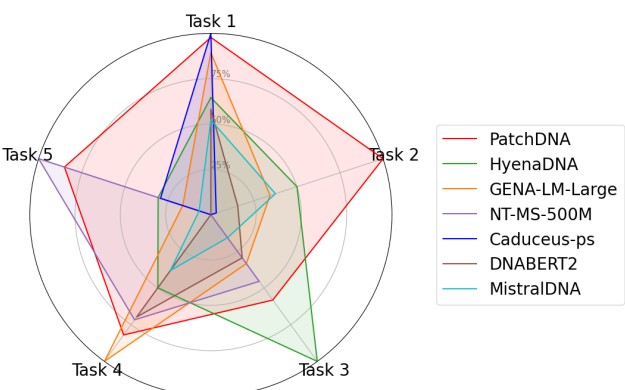

Figure 3: Radar plot showing normalized performance across all five tasks, with the best and worst performance for each task scaled to 1 and 0 respectively. Each axis corresponds to a different task, and larger enclosed area indicates stronger overall performance.

EXTENDED RESULTS

We present extended results for Tasks 3, 4 and 5 in Tables 13, 15, 14. In the main results in the paper, we report the Overall Accuracy for Task 3, the mean Spearman $r$ across the 5 cell types for Task 4, and the mean AUROC for Task 5.

Table 13: Accuracy and AUROC across different cell types for Task 3 in DART-Eval

| Model | Overall Accuracy | GM12878 | H1ESC | HEPG2 | IMR90 | K562 |
|---|---|---|---|---|---|---|
| PatchDNA | 0.457 | 0.740 | 0.817 | 0.806 | 0.783 | 0.710 |
| Caduceus | 0.281 | 0.535 | 0.622 | 0.680 | 0.576 | 0.587 |
| DNABERT2 | 0.371 | 0.652 | 0.757 | 0.762 | 0.691 | 0.691 |
| GENA-LM-Large | 0.383 | 0.627 | 0.787 | 0.773 | 0.714 | 0.693 |
| HyenaDNA | **0.587** | **0.849** | **0.889** | **0.862** | **0.882** | **0.799** |
| Mistral-DNA | 0.329 | 0.582 | 0.678 | 0.723 | 0.643 | 0.646 |
| NT-MS-500M | 0.420 | 0.744 | 0.795 | 0.783 | 0.779 | 0.711 |

Table 14: Zero-shot AUROC performance using embedding-based predictions for African and Yoruban datasets for Task 5 in DART-Eval

| Model | African AUROC | Yoruban AUROC |
|---|---|---|
| PatchDNA | **0.545** | 0.564 |
| Caduceus | 0.519 | 0.508 |
| DNABERT2 | 0.480 | 0.505 |
| GENA-LM-Large | 0.508 | 0.501 |
| HyenaDNA | 0.515 | 0.515 |
| Mistral-DNA | 0.520 | 0.475 |
| NT-MS-500M | 0.519 | **0.613** |

Table 15: Spearman $r$ among positives across five cell types for Task 4 in DART-Eval

| Model | GM12878 | H1ESC | HEPG2 | IMR90 | K562 |
|---|---|---|---|---|---|
| PatchDNA | 0.434 | 0.636 | 0.400 | 0.319 | 0.412 |
| Caduceus | 0.251 | 0.371 | 0.312 | 0.149 | 0.401 |
| DNABERT2 | 0.395 | 0.584 | 0.357 | 0.275 | 0.483 |
| GENA-LM-Large | **0.490** | **0.678** | **0.401** | **0.329** | 0.461 |
| HyenaDNA | 0.362 | 0.538 | 0.345 | 0.237 | 0.438 |
| Mistral-DNA | 0.293 | 0.500 | 0.349 | 0.244 | 0.431 |
| NT-MS-500M | 0.410 | 0.595 | 0.337 | 0.270 | **0.499** |

### A.3.3 BEND BENCHMARK

This section reports on the remaining BEND benchmark tasks(Marin et al., 2024). Table 2 focuses on short- and medium-range tasks, whereas the BEND enhancer task requires 100 kbp inputs. We evaluate this task only for models that can process such long sequences in a single forward pass. Specifically, we compare PatchDNA-7M (131 kbp context) with HyenaDNA Large and Caduceus-ph (Table 16). The enhancer task is a binary classification task that predicts whether a 128 bp region lies within an enhancer, using a 100 kbp surrounding context.

We additionally report zero-shot performance on the BEND Variant Effect Prediction (VEP) tasks, see Table 17. As highlighted by Dart-Eval (Patel et al., 2024), these tasks do not account for linkage disequilibrium, leading to potentially noisy labels and reduced reliability. For this reason, we include the results for completeness but exclude them from the primary comparisons.

Table 16: Enhancer annotation task performance. These results compare models capable of handling 100kbp sequence length inputs in a single pass. We report AUPRC across 10 cross validation folds.

| Model | AUPRC |
|---|---|
| PatchDNA (131k) | **0.037 $\pm$ 0.026** |
| HyenaDNA Large | 0.031 $\pm$ 0.019 |
| Caduceus-ph | 0.032 $\pm$ 0.020 |

Table 17: Zero-shot variant effect prediction (VEP) performance in terms of AUROC.

| Model | eQTL AUROC | Disease AUROC |
|---|---|---|
| PatchDNA | 0.49 | **0.82** |
| HyenaDNA | **0.51** | 0.45 |
| GENA-LM Large | 0.49 | 0.55 |
| NT-MS-500M | 0.48 | 0.48 |
| DNABERT-2 | 0.49 | 0.51 |

A.3.4 CAGE PREDICTION BENCHMARK

We use the CAGE dataset from https://huggingface.co/datasets/InstaDeepAI/genomics-long-range-benchmark, consisting of 50 CAGE tracks selected from the original 638 in the Basenji dataset.

Each model receives a sequence of 114,688 single nucleotides. We extract per-nucleotide embeddings and pass them through a two-layer MLP, where the hidden dimension is set to twice the embedding size and the output dimension is 50, following the setup in Brixi et al. (2025). The MLP outputs are mean-pooled over non-overlapping windows of 128 nucleotides, resulting in a final output of shape 896×50.

Training is performed using the Poisson negative log-likelihood loss, as in Enformer (Avsec et al., 2021). We fully finetune each model for one epoch, consistent with Brixi et al. (2025). We use the Adam optimizer, with a learning rate of $5e-5$ and a total batch size of 8.

For baseline models, `HyenaDNA`, `Caduceus-ps` and `Caduceus-ph` we use the pretrained weights available via Hugging Face, with model identifiers listed in Table 6.

For regulatory element based patching, we use annotations from Moore et al. (2020), creating a score function, $g_p$, that assigns a value of 1 to nucleotides in these regions, and 0 otherwise. We then use a patching threshold, $\theta_p$, of 0.99.

All experiments are repeated with five random seeds. We report the mean and standard deviation of performance on the test set, using the same metrics as Avsec et al. (2021), described in Section A.10. Finetuning runtimes for one epoch are reported in Table 18.

Table 18: One epoch finetuning time and FLOPS for various models, using 4 A100 80GB GPUs on CAGE prediction benchmark. The peak VRAM usage is normalised by batch size.

| Model | Time (minutes) | FWD FLOPS (G) | Peak VRAM usage (GB) |
|---|---|---|---|
| PatchDNA | 22.4 | 678.60 | 14.1 |
| HyenaDNA | 76.6 | 1493.96 | 20.4 |
| Caduceus-ph | 99.2 | 3142.71 | 18.1 |
| Caduceus-ps | 238.3 | 6285.42 | 36.2 |

A.3.5 CELL TYPE SPECIFIC RE-PATCHING

We pick paired CAGE-DNase tracks from the Basenji dataset (Kelley, 2020), focusing on Neurons, Hepatocytes and K562. The ids for the tracks that we used are in Table 19. We keep the same train/validation/test splits. For each cell type we follow the same protocol outlined in Section A.3.4, where instead of predicting 50 tracks we predict only 1 track. Since only 1 track is predicted, we opt to focus on cell correlation.

**DNase patching details** The DNase-seq data used for patching were obtained from the ENCODE Project portal (https://www.encodeproject.org/) using the ENCODE ids in Table 19. We use a patching threshold, $\theta_p$, of 0.99 for all DNase sources.

Table 19: Dataset identifiers for paired DNase-seq and CAGE expression tracks used in the cell-type-specific prediction task.

| Cell Type | DNase ENCODE ID | CAGE FANTOM5 ID |
|---|---|---|
| K562 | ENCFF413AHU | CNhs11250 |
| Hepatocyte | ENCFF136YOJ | CNhs12338 |
| Neuron | ENCFF399ISP | CNhs12338 |

A.3.6 THE GENOMICS LONG RANGE BENCHMARK

To evaluate performance on additional long-range prediction tasks, we use the Genomics Long Range Benchmark (Trop et al., 2025). We use a sequence length of 131 kbp, and restrict comparisons to architectures capable of processing this full context in a single forward pass, namely

HyenaDNA and Caduceus-ph (Caduceus-ps is excluded as it is much slower to run, see Table 18). Following the benchmark protocol, we adopt the authors' recommended hyperparameters and report the mean performance across five random seeds on the designated held-out test set (Table 20). Across these tasks, PatchDNA-7M delivers competitive or superior results, outperforming baseline models on 6 out of 7 tasks.

We further show results on the zeroshot tasks in Table 21. We use the same protocol for extracting zero shot scores as detailed in (Trop et al., 2025) for masked language models (Caduceus-ph), and autoregressive models (PatchDNA, HyenaDNA). On Pathogenic Clinvar, our model outperforms HyenaDNA and Caduceus-ph. On the Causal eQTL and OMIM tasks, all models perform close to random, consistent with the reported difficulties that DNA language models have in the zero shot setting on these tasks (Trop et al., 2025).

Table 20: Performance on Genomics Long Range Benchmark on all finetuning tasks. Results are reported across 5 seeds on the held out test set.

| Model | Causal eQTL | Pathogenic ClinVar | Bulk RNA | Histone Marks | DNA Accessibility | Promoter | Enhancer |
|---|---|---|---|---|---|---|---|
| | AUROC | AUROC | $R^2$ | AUPRC | AUPRC | AUPRC | AUROC |
| PatchDNA-7M | $0.714 \pm 0.005$ | $\mathbf{0.796 \pm 0.009}$ | $\mathbf{0.500 \pm 0.005}$ | $\mathbf{0.289 \pm 0.005}$ | $\mathbf{0.179 \pm 0.012}$ | $\mathbf{0.783 \pm 0.021}$ | $\mathbf{0.835 \pm 0.002}$ |
| HyenaDNA | $0.715 \pm 0.003$ | $0.622 \pm 0.044$ | $0.458 \pm 0.002$ | $0.245 \pm 0.006$ | $0.106 \pm 0.002$ | $0.704 \pm 0.025$ | $0.782 \pm 0.037$ |
| Caduceus-ph | $\mathbf{0.717 \pm 0.010}$ | $0.699 \pm 0.010$ | $0.491 \pm 0.014$ | $0.215 \pm 0.004$ | $0.118 \pm 0.005$ | $0.734 \pm 0.025$ | $0.822 \pm 0.002$ |

Table 21: Performance on Genomics Long Range Benchmark zeroshot tasks.

| Model | Causal eQTL | Pathogenic ClinVar | OMIM |
|---|---|---|---|
| | AUROC | AUROC | AUPRC |
| PatchDNA-7M | 0.487 | **0.586** | **0.00208** |
| Caduceus-ph | 0.479 | 0.501 | 0.00177 |
| HyenaDNA | 0.481 | 0.494 | 0.00187 |

## A.4 ABLATIONS

We present ablations to assess (i) how a conservation score only baseline performs, (ii) the effectiveness of conservation-based patching versus entropy and fixed size patching, and (iii) the contribution of patching and the BLT architecture itself.

`PatchDNA-Entropy` and `PatchDNA-FixedPS20` are pretrained and evaluated with entropy- and fixed- patching at matched efficiency to conservation-based patching (see Section A.2 for hyperparameters). For reference, we also include NT-MS-500M, the largest baseline in our benchmarks (500M parameters, multi-species).

We also construct a PhyloP baseline. For sequence-level tasks, PhyloP scores are pooled across the sequence by summation. For binary classification, we report AUROC or MCC depending on the established metric in literature for the benchmark. In the case of MCC, we fit a small linear probe to the scores to obtain binary predictions. For multiclass classification, we train a probe on PhyloP scores. For regression, we report direct correlation between scores and labels. For variant effect prediction, we follow Brixi et al. (2025) by taking the PhyloP score at the variant site and computing AUROC against effect/no-effect labels.

Across benchmarks, conservation-based patching outperforms entropy and fixed patching, highlighting the advantage of a biologically informed patching strategy. Furthermore, entropy and fixed patching remain strong competitors in comparison to NT-MS-500M, highlighting the strength of patching and the BLT architecture itself in DNA modeling. We also directly examine the relationship between conservation scores and task labels. We find that conservation scores alone cannot reproduce model performance. In most tasks, there is weak or no correlation to labels, and `PatchDNA` still outperforms baselines. In tasks where conservation is more predictive, conservation based patching achieves substantial gains over the PhyloP baseline.

Table 22: Performance across NT benchmark, with a linear probe on top of model embeddings. All results are using MCC. PhyloP baseline uses a linear probe.

| Task | PatchDNA | PatchDNA-Entropy | PatchDNA-FixedPS20 | NT-MS-500M | PhyloP |
|---|---|---|---|---|---|
| H2AFZ | $0.396 \pm 0.005$ | $0.401 \pm 0.007$ | $\textbf{0.405} \pm \textbf{0.005}$ | $0.392 \pm 0.005$ | $-0.017 \pm 0.062$ |
| H3K27ac | $\textbf{0.410} \pm \textbf{0.022}$ | $0.352 \pm 0.008$ | $0.386 \pm 0.006$ | $0.398 \pm 0.004$ | $0.105 \pm 0.028$ |
| H3K27me3 | $\textbf{0.557} \pm \textbf{0.004}$ | $0.529 \pm 0.004$ | $0.522 \pm 0.008$ | $0.536 \pm 0.004$ | $0.233 \pm 0.032$ |
| H3K36me3 | $\textbf{0.542} \pm \textbf{0.004}$ | $0.498 \pm 0.006$ | $0.486 \pm 0.017$ | $0.496 \pm 0.006$ | $0.304 \pm 0.003$ |
| H3K4me1 | $\textbf{0.406} \pm \textbf{0.009}$ | $0.381 \pm 0.009$ | $0.392 \pm 0.005$ | $0.391 \pm 0.009$ | $0.006 \pm 0.041$ |
| H3K4me2 | $0.459 \pm 0.004$ | $0.457 \pm 0.013$ | $0.469 \pm 0.012$ | $\textbf{0.470} \pm \textbf{0.005}$ | $-0.020 \pm 0.058$ |
| H3K4me3 | $0.614 \pm 0.006$ | $0.583 \pm 0.006$ | $0.592 \pm 0.010$ | $\textbf{0.622} \pm \textbf{0.007}$ | $0.009 \pm 0.084$ |
| H3K9ac | $0.470 \pm 0.011$ | $0.458 \pm 0.023$ | $0.486 \pm 0.015$ | $\textbf{0.514} \pm \textbf{0.005}$ | $0.026 \pm 0.041$ |
| H3K9me3 | $\textbf{0.393} \pm \textbf{0.012}$ | $0.346 \pm 0.009$ | $0.350 \pm 0.013$ | $0.304 \pm 0.016$ | $0.072 \pm 0.035$ |
| H4K20me1 | $\textbf{0.576} \pm \textbf{0.008}$ | $0.554 \pm 0.005$ | $0.563 \pm 0.005$ | $0.561 \pm 0.001$ | $0.059 \pm 0.103$ |
| enhancers | $0.475 \pm 0.004$ | $0.454 \pm 0.010$ | $0.448 \pm 0.013$ | $\textbf{0.485} \pm \textbf{0.003}$ | $0.181 \pm 0.007$ |
| enhancers_types | $0.441 \pm 0.005$ | $0.421 \pm 0.008$ | $0.413 \pm 0.014$ | $\textbf{0.445} \pm \textbf{0.003}$ | $0.167 \pm 0.002$ |
| promoter_all | $\textbf{0.779} \pm \textbf{0.007}$ | $0.719 \pm 0.007$ | $0.719 \pm 0.005$ | $0.718 \pm 0.003$ | $0.405 \pm 0.002$ |
| promoter_no_tata | $\textbf{0.786} \pm \textbf{0.003}$ | $0.743 \pm 0.003$ | $0.751 \pm 0.009$ | $0.741 \pm 0.004$ | $0.393 \pm 0.006$ |
| promoter_tata | $\textbf{0.853} \pm \textbf{0.009}$ | $0.749 \pm 0.040$ | $0.765 \pm 0.018$ | $0.685 \pm 0.032$ | $0.469 \pm 0.006$ |
| splice_sites_acceptors | $\textbf{0.669} \pm \textbf{0.006}$ | $0.497 \pm 0.005$ | $0.512 \pm 0.012$ | $0.468 \pm 0.005$ | $0.543 \pm 0.001$ |
| splice_sites_all | $\textbf{0.454} \pm \textbf{0.018}$ | $0.311 \pm 0.011$ | $0.310 \pm 0.013$ | $0.336 \pm 0.005$ | $0.283 \pm 0.004$ |
| splice_sites_donors | $\textbf{0.692} \pm \textbf{0.014}$ | $0.512 \pm 0.007$ | $0.521 \pm 0.019$ | $0.509 \pm 0.004$ | $0.547 \pm 0.001$ |

Table 23: Performance on DART-Eval. Task 3 is a 5 way classification task, where random performance is approximately 0.200. Task 4 is a regression task, Task 5 is a variant effect prediction task.

| Model | Task 1 | Task 2 | Task 3 | Task 4 | Task 5 |
|---|---|---|---|---|---|
| | Accuracy | Accuracy | Accuracy | Spearman $R$ | AUROC |
| PatchDNA | 0.966 | **0.725** | 0.457 | **0.440** | 0.555 |
| PatchDNA-FixedPS20 | **0.967** | 0.675 | **0.477** | 0.417 | 0.539 |
| PatchDNA-Entropy | 0.965 | 0.650 | 0.465 | 0.400 | 0.523 |
| NT-MS-500M | 0.745 | 0.565 | 0.420 | 0.422 | **0.566** |
| PhyloP | N/A | N/A | 0.260 | 0.027 | 0.536 |

Table 24: Performance across BEND short and long range tasks. Gene finding is a multi class classification task, reported with MCC, while other tasks are binary classification.

| Model | Gene finding | Chromatin accessibility | Histone modification | CpG Methylation |
|---|---|---|---|---|
| | MCC | AUROC | AUROC | AUROC |
| PatchDNA | 0.58 | **0.84** | **0.79** | **0.92** |
| PatchDNA-FixedPS20 | 0.38 | 0.83 | 0.78 | 0.90 |
| PatchDNA-Entropy | 0.37 | 0.83 | 0.78 | 0.90 |
| NT-MS-500M | **0.64** | 0.80 | 0.76 | 0.91 |
| PhyloP | 0.19 | 0.54 | 0.51 | 0.49 |

## A.5 SCALING PATCHDNA TO MULTIPLE SPECIES AND TO MULTIPLE GENOMES WITHIN THE SAME SPECIES

PhyloP scores quantify conservation at each genomic position using multi-species alignments. By leveraging evolutionary constraints, biologically relevant indicators of functional importance across species, the conservation-based patching approach is conceptually robust. Since PhyloP tracks are available for many organisms and genome assemblies, extending our model to a multi-species framework is straightforward.

We constructed a mouse version of the CAGE prediction task, by selecting 50 mouse CAGE tracks from the Basenji dataset. We applied PatchDNA using PhyloP conservation scores from the 60-way multi-species alignment for mouse (`mm10.60way.phyloP60way.bw`). The setup was the same as in Section 4.4: all models were fine-tuned for one epoch using an MLP head. Due to time constraints, we report only the Full Pearson correlation between predicted and observed CAGE signal across gene-cell pairs (computing gene-wise and cell-wise correlations required incorporating

transcription start site annotations for the mouse genome, which we plan to include in the final version). Despite being trained on the human genome, PatchDNA achieves strong performance on this task, outperforming HyenaDNA. This result highlights PatchDNA's ability to generalize across species, leveraging evolutionary priors without retraining.

Results are reported on the test set, averaged across 6 random seeds.

| Model | Full Pearson |
|---|---|
| HyenaDNA-1m-seqlen | $0.219 \pm 0.004$ |
| PatchDNA-7M | $0.338 \pm 0.004$ |

Table 25: Mouse CAGE prediction results using conservation-based PatchDNA.

## A.6 Pseudocode for re-patching

We also provide a simplified algorithm for establishing the patch boundaries below:

---

**Algorithm 1** DetectPatchBoundaries

---

**Require:** Input byte sequence $input$ of length $L$; genome scores $genome\_scores$ of length $L$; threshold $\tau$
**Ensure:** List $patch\_boundaries$
 1: Initialize empty list $patch\_boundaries \leftarrow []$
 2: **append-front** 1 to $patch\_boundaries$
 3: **append-front** 0 to $patch\_boundaries$
 4: **for** $i \leftarrow 0$ **to** $L - 1$ **do**
 5:    **if** $genome\_scores[i] > \tau$ **then**
 6:       **append** $i$ to $patch\_boundaries$
 7:    **end if**
 8: **end for**
 9: **return** $patch\_boundaries$

---

Integrating re-patching is straightforward. The `patchDNA` backbone accepts a `patching_mode` argument specifying the patching strategy, which dynamically defines the patch boundaries. These boundaries are used by the local encoder and decoders to determine how patches interact via cross-attention. This method is entirely data driven and does not require retraining. Below is a simple example:

```
model = PatchDNA.load_checkpoint("best.ckpt") # Trained with PhyloP
data_cfg = {
  "genome_score_fn": "dnase_k562", # instead of phylop
   # other cfg items
   }
dataset = Dataset.from_config(data_cfg)
model.architecture.patcher.threshold = 0.99
model.architecture.patcher.patching_mode = "custom_genome_scores"

# inference or finetune loop ...
preds = trainer.validate(model, datamodule)
```

## A.7 Alternative conservation scores and sensitivity to thresholds

PhastCons is an alternative conservation scoring method, but we deprioritized using it due to its window-based smoothing which results in lack of single nucleotide granularity. We present results in Table 27, showing that it underperforms compared to PhyloP on 3 out of the 4 tasks.

We pick the 95% threshold for efficiency reasons, as this allows us to easily train models at long sequences. Lower thresholds result in more number of patches, on average, increasing the computational cost. However, to investigate performance at other thresholds, we've run threshold-sensitivity

analyses for Dart-eval on the 7M-parameter PatchDNA using less stringent cutoffs (Task 4 was omitted due to increased computational costs). We highlight that performance does not change significantly between various thresholds (Table 26).

Table 26: Performance comparison of PatchDNA-7M variants on a subset of Dart-eval tasks

| Model | Avg. Patch Size | Task 1 | Task 2 | Task 3 | Task 5 |
|---|---|---|---|---|---|
| | | Accuracy | Accuracy | Accuracy | AUROC |
| PatchDNA-7M 75% | 4 | 0.938 | 0.645 | 0.343 | 0.524 |
| PatchDNA-7M 90% | 10 | 0.940 | 0.650 | 0.357 | 0.525 |
| PatchDNA-7M 95% | 20 | 0.950 | 0.650 | 0.380 | 0.539 |

Table 27: Performance comparison of PatchDNA-7M with PhastCon on a subset of Dart-eval tasks

| Model | Avg. Patch Size | Task 1 | Task 2 | Task 3 | Task 5 |
|---|---|---|---|---|---|
| | | Accuracy | Accuracy | Accuracy | AUROC |
| PatchDNA-7M 75% PhastCon | 4 | 0.882 | 0.615 | 0.332 | 0.534 |
| PatchDNA-7M 90% PhastCon | 10 | 0.932 | 0.645 | 0.326 | 0.542 |
| PatchDNA-7M 95% PhastCon | 20 | 0.943 | 0.640 | 0.333 | 0.549 |

We also investigate the impact of varying the threshold across a broad set of downstream tasks via re-patching. We use the main PatchDNA model, which has been trained using a PhyloP threshold which results in an average patch size of 20. We then identify thresholds corresponding to average patch sizes of 4, 10, 20, 40, 60, and 80, and evaluate the model with these alternative thresholds on both short-range and long-range tasks (Table 28 and Table 29).

In Table 28, we observe that smaller average patch sizes generally yield the best performance, although the improvements are modest for the regulatory element and chromatin profile tasks (Figure 4). For the splice-site tasks, the differences are more pronounced, which is expected given that these tasks benefit from finer-grained sequence resolution (see Table 12 in Section A.3.1).

A similar trend appears for the long-range CAGE task (Table 29). Performance declines gradually as the average patch size increases, but the drop is modest: even at the largest patch size of 80, PatchDNA still outperforms the second-best model in the benchmark (Caduceus-ps).

These results suggest that while finer patching can provide advantages, particularly for tasks requiring high-resolution sequence information, the model remains broadly robust across a wide range of patch sizes. Notably, smaller patch sizes incur higher computational cost, as they increase the number of patches that must be processed. In conventional tokenization schemes, modifying an analogous parameter (such as the k in k-mer tokenization) would necessitate training a new foundation model from scratch. In contrast, our framework enables users to re-patch post hoc to achieve smaller effective patch sizes, avoiding the substantial computational expense of pretraining a new model.

Table 28: Results on the finetuned Nucleotide Transformer benchmark using a PatchDNA model pretrained with conservation scores and re-patched at different average patch sizes. Test MCC is shown and averaged across 3 seeds with reported standard deviations.

| Task | Average Patch Size | | | | | |
|---|---|---|---|---|---|---|
| | 4 | 10 | 20 | 40 | 60 | 80 |
| H2AFZ | $0.517 \pm 0.014$ | $0.511 \pm 0.001$ | $\mathbf{0.523} \pm 0.010$ | $0.515 \pm 0.014$ | $0.506 \pm 0.015$ | $0.519 \pm 0.009$ |
| H3K27ac | $\mathbf{0.526} \pm 0.017$ | $0.510 \pm 0.020$ | $0.486 \pm 0.015$ | $0.495 \pm 0.036$ | $0.487 \pm 0.045$ | $0.486 \pm 0.032$ |
| H3K27me3 | $\mathbf{0.616} \pm 0.011$ | $0.614 \pm 0.016$ | $0.607 \pm 0.008$ | $0.590 \pm 0.014$ | $0.588 \pm 0.026$ | $0.596 \pm 0.011$ |
| H3K36me3 | $\mathbf{0.632} \pm 0.006$ | $0.631 \pm 0.009$ | $0.621 \pm 0.007$ | $0.620 \pm 0.001$ | $0.607 \pm 0.002$ | $0.606 \pm 0.009$ |
| H3K4me1 | $\mathbf{0.489} \pm 0.004$ | $0.475 \pm 0.009$ | $0.480 \pm 0.003$ | $0.476 \pm 0.011$ | $0.474 \pm 0.002$ | $0.473 \pm 0.010$ |
| H3K4me2 | $0.570 \pm 0.003$ | $\mathbf{0.581} \pm 0.017$ | $0.573 \pm 0.004$ | $0.570 \pm 0.004$ | $0.570 \pm 0.007$ | $0.575 \pm 0.003$ |
| H3K4me3 | $\mathbf{0.641} \pm 0.016$ | $0.617 \pm 0.016$ | $0.634 \pm 0.005$ | $0.633 \pm 0.015$ | $0.613 \pm 0.022$ | $0.628 \pm 0.017$ |
| H3K9ac | $\mathbf{0.589} \pm 0.012$ | $0.572 \pm 0.007$ | $0.569 \pm 0.010$ | $0.565 \pm 0.011$ | $0.567 \pm 0.015$ | $0.556 \pm 0.007$ |
| H3K9me3 | $0.485 \pm 0.021$ | $0.480 \pm 0.018$ | $0.470 \pm 0.017$ | $\mathbf{0.495} \pm 0.039$ | $0.473 \pm 0.032$ | $0.475 \pm 0.027$ |
| H4K20me1 | $\mathbf{0.670} \pm 0.008$ | $0.650 \pm 0.006$ | $0.637 \pm 0.007$ | $0.635 \pm 0.003$ | $0.626 \pm 0.015$ | $0.627 \pm 0.009$ |
| enhancers | $\mathbf{0.554} \pm 0.005$ | $0.536 \pm 0.013$ | $0.528 \pm 0.009$ | $0.521 \pm 0.008$ | $0.532 \pm 0.001$ | $0.524 \pm 0.005$ |
| enhancers_types | $\mathbf{0.519} \pm 0.012$ | $0.501 \pm 0.014$ | $0.496 \pm 0.008$ | $0.484 \pm 0.021$ | $0.492 \pm 0.004$ | $0.497 \pm 0.007$ |
| promoter_all | $0.781 \pm 0.012$ | $\mathbf{0.792} \pm 0.012$ | $0.791 \pm 0.009$ | $0.781 \pm 0.003$ | $0.791 \pm 0.008$ | $0.783 \pm 0.010$ |
| promoter_no_tata | $\mathbf{0.797} \pm 0.008$ | $0.795 \pm 0.012$ | $0.788 \pm 0.005$ | $0.796 \pm 0.012$ | $0.794 \pm 0.006$ | $0.783 \pm 0.008$ |
| promoter_tata | $\mathbf{0.875} \pm 0.013$ | $0.829 \pm 0.050$ | $0.840 \pm 0.019$ | $0.847 \pm 0.024$ | $0.843 \pm 0.011$ | $0.830 \pm 0.010$ |
| splice_sites_acceptors | $\mathbf{0.868} \pm 0.029$ | $0.741 \pm 0.026$ | $0.754 \pm 0.040$ | $0.748 \pm 0.044$ | $0.778 \pm 0.040$ | $0.746 \pm 0.040$ |
| splice_sites_all | $\mathbf{0.849} \pm 0.004$ | $0.789 \pm 0.008$ | $0.760 \pm 0.019$ | $0.772 \pm 0.016$ | $0.803 \pm 0.083$ | $0.778 \pm 0.084$ |
| splice_sites_donors | $\mathbf{0.744} \pm 0.005$ | $0.721 \pm 0.024$ | $0.706 \pm 0.026$ | $0.714 \pm 0.029$ | $0.705 \pm 0.029$ | $0.690 \pm 0.012$ |

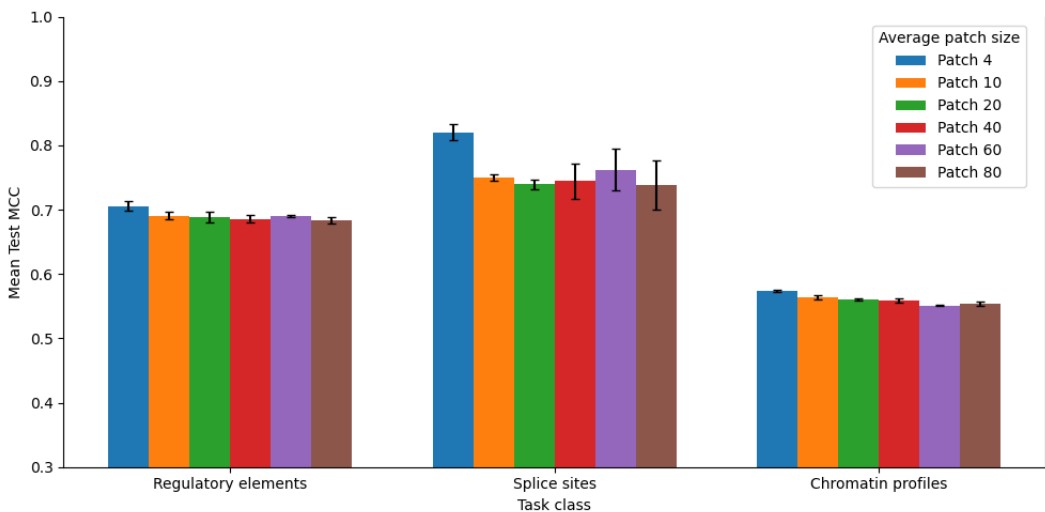

Figure 4: Results on finetuned Nucleotide Transformer benchmark, grouping tasks by category (Table 28). Using a PatchDNA model pretrained with conservation scores and re-patched at different average patch sizes

Table 29: Results on CAGE benchmark using a PatchDNA model pretrained with conservation scores and re-patched at different average patch sizes. Note that we exclude average patch size 4, as this would be inefficient at a long sequence of $< 100kbp$. Results are averaged across 5 seeds, with reported standard deviations.

| Average Patch Size | Gene Corr | Cell Corr | Full Pearson | Forward FLOPS |
|---|---|---|---|---|
| 10 | $\mathbf{0.369 \pm 0.001}$ | $\mathbf{0.773 \pm 0.0007}$ | $\mathbf{0.482 \pm 0.000}$ | 855.64 |
| 20 | $\mathbf{0.369 \pm 0.001}$ | $0.772 \pm 0.0018$ | $0.471 \pm 0.002$ | 678.55 |
| 40 | $0.368 \pm 0.000$ | $0.770 \pm 0.0015$ | $0.454 \pm 0.005$ | 627.88 |
| 60 | $0.367 \pm 0.002$ | $0.767 \pm 0.0005$ | $0.440 \pm 0.003$ | 616.61 |
| 80 | $0.367 \pm 0.002$ | $0.769 \pm 0.0018$ | $0.425 \pm 0.004$ | 612.02 |

## A.8 INTERPRETABILITY: QUANTITATIVE ANALYSIS OF PATCH ALIGNMENT WITH cCREs

We believe that interpretability, particularly the alignment of patches with known functional genomic elements, is important. To address this, we implemented an additional quantitative analysis comparing the enrichment of PhyloP-derived patches specifically within cCRE versus non-cCRE genomic regions. We used 5 independent random seeds, each with 5000 sampled genomic intervals of length 350 bp. For regulatory regions, we centered the windows on known cCREs (from ENCODE), while control intervals were drawn from the genome to avoid any overlap with cCRE annotations. Using a Mann–Whitney U test, we found that there were significantly more PhyloP-derived patches (median difference: 32, Cliff's $\delta = 0.618$, $p \ll 0.001$) within cCRE regions relative to randomly sampled non-cCRE genomic windows.

Further, we compared the number of patches identified by entropy and PhyloP scores within cCRE regions using the Wilcoxon signed-rank test. PhyloP-derived patches consistently identified significantly more patches per region than entropy-derived patches (median difference: 12 patches, Cliff's $\delta = 0.155$, Wilcoxon $p \ll 0.001$). While this effect is statistically robust across seeds, the effect sizes are smaller than those observed in the cCRE vs. control comparisons.

## A.9 COMPUTATIONAL EFFICIENCY OF PATCHING AND RE-PATCHING

The re-patching itself incurs no additional computational overhead: the local encoder and decoder already expect a patch-based layout, which can be swapped in without changing the architecture. The patch size distribution will have a direct effect on computations. The computational complexity of marking patch boundaries is an $\mathcal{O}(L)$ operation (with $L$ being the sequence length): we make a single pass over the sequence, inserting boundaries whenever a pre-established threshold is reached. In our implementation this step runs on the CPU, though an entropy-based patching strategy would necessitate executing a small model on the GPU and will have different computational complexity considerations. To clarify this further, we present the theoretical computational cost (in GFLOPs) in Table 30 comparing `PatchDNA` directly against its single-nucleotide baseline, where the patch size is fixed at 1. These theoretical estimates were calculated using the formulas described in the BLT paper, as the BLT implementation uses FlexAttention (which Pytorch FLOP profilers don't support).

Table 30: Forward FLOPs comparison across models at different sequence lengths.

| Model | 511 bp FWD FLOPS (G) | 16 kbp FWD FLOPS (G) |
|---|---|---|
| PatchDNA (19.2 M) | 5.64 | 179.07 |
| Single-nucleotide baseline (19.2 M) | 11.80 | 1384.53 |
| PatchDNA (7.7 M) | 2.79 | 88.6 |
| Single-nucleotide baseline (7.7 M) | 5.36 | 548.62 |

## A.10 METRICS

**Matthews Correlation Coefficient (MCC)** The Matthews Correlation Coefficient is a robust statistical rate which takes into account true and false positives and negatives and is regarded as a balanced measure that can be used even if the classes are of very different sizes.

$$\text{MCC} = \frac{TP \times TN - FP \times FN}{\sqrt{(TP + FP)(TP + FN)(TN + FP)(TN + FN)}}$$

where $TP$, $TN$, $FP$, and $FN$ are the numbers of true positives, true negatives, false positives, and false negatives, respectively.

### A.10.1 ENFORMER EVALUATION METRICS

To assess model performance in predicting gene expression, we follow Pearson correlation evaluation strategies as proposed in the Enformer manuscript (Avsec et al., 2021). The following three metrics are used to evaluate model predictions: gene correlation, cell correlation, and full correlation.

Let $\hat{W} \in \mathbb{R}^{B \times C}$ and $W \in \mathbb{R}^{B \times C}$ denote the predicted and observed CAGE matrices across the genome, where $B$ is the number of genomic bins (each spanning 128 base pairs) and $C$ is the number of cell types.

To obtain gene-level predictions, we extract the row of $\hat{W}$ and $W$ corresponding to the bin that contains the transcription start site (TSS) of each gene. This gives the predicted and observed gene expression matrices $\hat{Y}, Y \in \mathbb{R}^{G \times C}$, where $G$ is the number of genes.

**Gene Correlation**    Gene correlation evaluates how well the model captures cell type–specific expression patterns for each gene. Prior to computing this metric, both predicted and observed gene expression values are log-transformed as:

$$\hat{Y} \leftarrow \log(\hat{Y} + 1), \quad Y \leftarrow \log(Y + 1)$$

For each gene $g \in \{1, \ldots, G\}$, we compute the Pearson correlation across all cell types:

$$r_g^{\text{gene}} = \text{corr}(\hat{Y}_{g,:}, Y_{g,:})$$

The final gene correlation score is the average over all genes:

$$r^{\text{gene}} = \frac{1}{G} \sum_{g=1}^{G} r_g^{\text{gene}}$$

**Cell Correlation**    Cell correlation evaluates how well the model predicts gene expression patterns across genes within each cell type. As with gene correlation, a log-transformation is applied to all input values before computing correlation.

For each cell type $c \in \{1, \ldots, C\}$, we compute the Pearson correlation across genes:

$$r_c^{\text{cell}} = \text{corr}(\hat{Y}_{:,c}, Y_{:,c})$$

The final cell correlation score is the average over all cell types:

$$r^{\text{cell}} = \frac{1}{C} \sum_{c=1}^{C} r_c^{\text{cell}}$$

**Full Correlation**    Full correlation measures how well the model predicts CAGE signal profile across the genome.

For each cell type $c \in \{1, \ldots, C\}$, we compute the Pearson correlation across bins:

$$r_c^{\text{full}} = \text{corr}(\hat{W}_{:,c}, W_{:,c})$$

The final full correlation score is the average over all the cell types

$$r^{\text{full}} = \frac{1}{C} \sum_{c=1}^{C} r_c^{\text{full}}$$

## A.11    LLM USAGE

We have used LLMs to improve grammar and wording throughout the manuscript.

