# OpenReview forum: "PatchDNA: A Flexible and Biologically-Informed Alternative to Tokenization for DNA"
_ICLR.cc/2026/Conference — ICLR 2026 Poster_

### Official Review · Reviewer_3uWp · 2025-10-31

**Soundness:** 2
**Presentation:** 2
**Contribution:** 2
**Rating:** 2
**Confidence:** 5

**Summary:**

This paper proposes PatchDNA, a patch‐based alternative to tokenization for DNA sequence modeling. PatchDNA segments the nucleotide sequence into variable-length patches determined by a scoring function.

**Strengths:**

1. The local–global–local stack is well matched to genomics: local attention preserves base-level detail and global attention over patches amortizes long-range reasoning.

2. The paper conducts a broad set of ablation studies, which makes the empirical story more credible and helps clarify what actually drives the gains.

**Weaknesses:**

1. An advantage is retaining single-base resolution, yet there is limited coverage of base-level benchmarks where this matters most, variant effect prediction (VEP) .

2. The method is positioned as long-context friendly, but long-range evaluation remains thin and not fully controlled. Missing or underdeveloped: Bend or LRB benchmark.


3. Ablations need to show how average patch length affects long-range performance.


4. Claiming “conservation as boundary is better than conservation as feature” lacks causal evidence and visualization. The paper asserts a strong conclusion without systematic attribution. It is unclear why boundaries confer benefits beyond adding conservation as an auxiliary input.

5. Need to visualize how boundary placement changes long-range motif cooperation.

6. Despite the elegant design, the method often trails or only matches strong baselines on core tasks, suggesting modest practical effectiveness until stronger results are shown under matched settings.

**Questions:**

Please see Weaknesses.

---

> ### Author Response · Authors · 2025-11-14
> **Request for clarification**
>
> Thank you for your review, before we write a full response to your points, we would like to ask some clarifying questions:
>
> > 5. Need to visualize how boundary placement changes long-range motif cooperation.
>
> We would appreciate a bit more clarity on the type of analysis you have in mind for visualizing how boundary placement influences long-range motif cooperation. If possible, are you able to elaborate with an example, or point us to a paper which analyses long range motif cooperation in DNA models, so that we can understand the kind of analysis you are looking for?
>
> > 6. Despite the elegant design, the method often trails or only matches strong baselines on core tasks, suggesting modest practical effectiveness until stronger results are shown under matched settings.
>
> Can you clarify what specific ‘matched settings’ you believe are missing in our comparisons?

---

> ### Author Response · Authors · 2025-11-20
> **Response (1/2)**
>
> We thank the reviewer for their feedback, and we address the raised concerns below:
>
> >An advantage is retaining single-base resolution, yet there is limited coverage of base-level benchmarks where this matters most, variant effect prediction (VEP) .
>
> We appreciate the reviewer’s point about the importance of base-level evaluation. The paper already includes single-base resolution tasks, such as the gene-finding task in BEND and the variant effect prediction task in Dart Eval, where PatchDNA is surpassed only by the largest model (NT-MS-500M).
>
> To further strengthen this aspect, we have added results on variant effect prediction from the Genomics Long Range Benchmark in Section A.3.6 of the revised manuscript. On disease variant prediction, PatchDNA outperforms the baseline DNA language models by a substantial margin. For eQTL prediction, PatchDNA performs on par with models that explicitly operate at single-nucleotide resolution (HyenaDNA and Caduceus-ph). These results collectively demonstrate that PatchDNA retains strong single-base sensitivity despite operating over patches.
>
> >The method is positioned as long-context friendly, but long-range evaluation remains thin and not fully controlled. Missing or underdeveloped: Bend or LRB benchmark.
>
> We describe PatchDNA as long-context friendly because it can process sequences exceeding 100 kbp efficiently, something that most existing DNA language models cannot achieve. Patch-based processing enables PatchDNA to handle these lengths faster and more flexibly than other long range models such as HyenaDNA or Caduceus.
>
> In response to the reviewer’s suggestion, we have expanded our long-range evaluation by adding the full BEND benchmark as well as additional tasks from the Genomics Long Range Benchmark. These results are included in Sections A.3.3 and A.3.6 of the revised manuscript. For fair comparison, we evaluate against models capable of handling \>100 kbp sequences in a single forward pass (HyenaDNA and Caduceus-ph). We do not run Caduceus-ps due to its significantly higher computational cost, approximately twice as slow as Caduceus-ph.
>
> **BEND results:**
>
> In the revised manuscript, we clarify that some of the variant effect prediction tasks in BEND are not ideal as core evaluations because they do not account for linkage disequilibrium among variants (as discussed in Dart-Eval \[1\]). Nevertheless, for completeness, we now include these results in Section A.3.3.
>
> We further report performance on the Enhancer task in Table 16 Section A.3.3 of the paper. The model performs comparably to HyenaDNA and Caduceus, whilst being much faster and more efficient (See Table 18 in Section A.3.4).
>
> **The Genomics Long Range Benchmark:**
>
> We additionally expand our long-context evaluation by including five tasks from the Genomics Long Range Benchmark, reported in Table 20 (Section A.3.6). Due to time and compute constraints, we omit the Promoter and Enhancer tasks.
>
> The model performs well across these new tasks, in particular we outperform HyenaDNA and Caduceus-ph on 4 out of 5 tasks. We believe that the expanded set of experiments in the revised manuscript now provides comprehensive coverage of the available evaluations.
>
> \[1\] Patel, A., Singhal, A., Wang, A., Pampari, A., Kasowski, M., & Kundaje, A. (2024). DART-Eval: A comprehensive DNA language model evaluation benchmark on regulatory DNA. The Thirty-eighth Conference on Neural Information Processing Systems: Datasets and Benchmarks Track

---

> ### Author Response · Authors · 2025-11-20
> **Response (2/2)**
>
> >Ablations need to show how average patch length affects long-range performance.
>
> Thank you for raising this point. To clarify, in the main paper we always use an average patch length of 20, based on our computational resources, and do not tune this hyperaparameter. However, to address the reviewer’s comment, in the revised paper, we include an ablation on the CAGE task covering a wide range of average patch sizes (Table 28 in Section A.7 [Edit: Table 29 in Section A.7 in latest version of the paper]). While smaller patch sizes yield slightly better performance, the degradation at larger sizes is minimal, and the model still outperforms the second-best benchmark model even at large average patch sizes (80).
>
> >Claiming “conservation as boundary is better than conservation as feature” lacks causal evidence and visualization. The paper asserts a strong conclusion without systematic attribution. It is unclear why boundaries confer benefits beyond adding conservation as an auxiliary input.
>
> We would like to clarify that we do not claim that conservation boundaries are strictly superior to adding conservation as an extra input feature to a model.
>
> Our goal is instead to demonstrate that patching offers an efficient and flexible mechanism for allocating computation, and conservation-based boundaries represent one practical instantiation of this idea. Using conservation to define patch boundaries highlights informative regions without increasing input dimensionality or requiring additional input channels, which is particularly advantageous when scaling to long sequences where standard tokenization becomes costly.
>
> Importantly, PatchDNA is agnostic to the choice of scoring function. Conservation is not a requirement: users can re-patch using entropy, fixed-size windows, task-specific scoring functions, or any future biologically motivated signal. In contrast, incorporating conservation as an explicit feature always requires having conservation tracks available at inference time, which may not hold across species, assemblies, or experimental contexts. Our intention is to illustrate that patch-based processing offers a more flexible and scalable alternative for leveraging conservation when appropriate.
>
> >Despite the elegant design, the method often trails or only matches strong baselines on core tasks, suggesting modest practical effectiveness until stronger results are shown under matched settings.
>
> We respectfully disagree with the reviewer’s assertion that PatchDNA “trails or only matches strong baselines on core tasks”. Across all benchmarks included in the revised manuscript, PatchDNA demonstrates consistently strong and often state-of-the-art performance. Note that our model is much faster than competitor models at similar sequence lengths, being x3.4 faster than HyenaDNA (Table 18 in Section A.3.4), providing practical benefits beyond accuracy alone.
>
> **Nucleotide Transformer benchmark:** On the three core groups of tasks, we outperform the baselines (including models which are much larger than ours) on 2 out of the 3 core groups, and we are second best in the final group (Chromatin profiles).
>
> **Dart-Eval:** We have the highest mean rank across the tasks, showing that the model is the most general learner. We provide a graph in Supplementary Section A.3.2 to further emphasise this point, which highlights that in Dart-Eval, whilst the model is not the best on every task (and no model is), it clearly has the most general performance.
>
> **BEND:** On the 4 short to medium range tasks, we are best in 3 out of 4 tasks (matching or beating baselines), and second best in Gene finding, only being outperformed by the largest model. In contrast, no other model is best in more than 1 task.
>
> **Long range tasks:** We outperform HyenaDNA and Caduceus on the CAGE task. In the extra Genomics Long Range benchmark results, we outperform matched baselines in 4 out of 5 tasks.
>
> Taken together, we believe the results demonstrate that PatchDNA is broadly the most performant model compared to the baselines, but also offers meaningful advantages in generality and efficiency.

---

> > ### Comment · Reviewer_3uWp · 2025-11-28
> >
> > Thanks for the detailed response, added benchmarks (BEND + Long Range Benchmark), and the ablations (patch size on CAGE). These address several of my earlier concerns, and I will increase my overall score.
> > I still have a couple of questions:
> >
> > 1. Why such a divergence between the two variant tasks in Table 17? PatchDNA appears merely competitive on the eQTL task but reaches SOTA on the Disease variant task (and by a large margin over many methods).
> >
> > 2. How about the performance with the zero-shot setting on the long-range benchmark’s variant tasks?

---

> > > ### Author Response · Authors · 2025-12-03
> > > **Response to further questions**
> > >
> > > We thank the reviewer for their response, and provide extra clarifications below:
> > >
> > > > Why such a divergence between the two variant tasks in Table 17? PatchDNA appears merely competitive on the eQTL task but reaches SOTA on the Disease variant task (and by a large margin over many methods).
> > >
> > > We thank the reviewer for highlighting the performance divergence between the Variant Effect Prediction (VEP) tasks in Table 17\. We believe this difference highlights the specific impact of our conservation-guided patching strategy. To verify this, we have now compared our method against the **PatchDNA-Entropy** model, which uses entropy based patching. For the disease task, this achieves 0.538, whereas conservation patching achieves 0.82, highlighting that performance gains are due to conservation driven patching. This confirms that the model is successfully leveraging a strong biological link between evolutionary conservation and pathogenicity.
> > >
> > > However, we note that as mentioned previously, the variant tasks in BEND may not account for  linkage disequilibrium, as pointed out by \[1\], which could affect the insights derived from these results.
> > >
> > > \[1\] Patel, A., Singhal, A., Wang, A., Pampari, A., Kasowski, M., & Kundaje, A. (2024). DART-Eval: A comprehensive DNA language model evaluation benchmark on regulatory DNA. The Thirty-eighth Conference on Neural Information Processing Systems: Datasets and Benchmarks Track
> > >
> > > >How about the performance with the zero-shot setting on the long-range benchmark’s variant tasks?
> > >
> > > We now include zero-shot results on the long-range variant tasks in the revised manuscript in Section A.3.6, Table 21\. In the disease variant prediction task (Clinvar), the model is performing better than the other models, which is consistent with the trend we observed in the BEND benchmark. For the other variant tasks, all models perform near random, as these are difficult tasks for DNA language models in the zero-shot setting.
> > >
> > > We have now also updated the paper with the remaining Enhancer and Promoter tasks for the Genomics LRB benchmark. The paper now includes the full set of finetuning tasks from the benchmark (Section A.3.6, Table 20). Our model outperforms Caduceus and HyenaDNA on 6 out of 7 tasks.

---

### Official Review · Reviewer_6uT1 · 2025-11-01

**Soundness:** 3
**Presentation:** 4
**Contribution:** 3
**Rating:** 8
**Confidence:** 3

**Summary:**

The paper proposes PatchDNA, a DNA language modeling framework that replaces traditional tokenization with a dynamic, vocabulary-free patching mechanism inspired by the Byte Latent Transformer. Patches are determined by biologically informed scoring functions such as evolutionary conservation, allowing the model to focus computational resources on functionally relevant regions. The framework further introduces re-patching, enabling modification of patch boundaries after pretraining for task or tissue specific adaptation without retraining. Experiments across multiple genomic benchmarks demonstrate improved efficiency, and competitive or superior performance compared to much reported baselines.

**Strengths:**

- Innovative representation: Replacing fixed tokenization with dynamic, vocabulary-free patching tailored to DNA sequences is innovative and is a domain (biology) inspired approach.

- Biological grounding: The proposed approach and biological prior grounded. Conservation-guided patch boundaries align model focus with functionally relevant genomic regions.

- Re-patching flexibility: The approach enables changing segmentation (in downstream/post-training) for new cell type or tissue contexts without the need of retraining. This makes is a practically useful one.

- Efficiency: The proposed method has reported to be cable of handles very large context (>100kb) contexts with small models. THis would reduce FLOPs compared to single-nucleotide models.

- Empirical breadth: The authors evaluated PatchDNA on four major genomic benchmarks where the PatchDNA achieve SoTA or near-SoTA results.

- Interpretability: Patch boundaries and conservation scores offer intuitive biological interpretability of learned representations.

- Technical rigor: Clear formulation, hyperparameters (essential for reproducibility), details of the baselines are provided.

**Weaknesses:**

- Limited testing: I would suggest the author conduct in-silico mutagenesis or perturbation analysis to verify that patches capture causal biological signals. Conducting in-silico mutagenesis is often not that expensive to run with limited computing resources.

- Statistical significance testing: Significance testing or variance reporting is missing. Adding this will clarify the performance gains.

- [Minor] Evaluation paradigms: All the benchmarks are supervised only. It would be interesting to see the PatchDNA's performance on unsupervised or generative task validation.

**Questions:**

- Can the authors conduct in-silico mutagenesis to confirm that patch boundaries capture causal biological signals?

- Can the authors report statistical significance for benchmark results and explore (or provide discussion on the potential of) PatchDNA’s performance on unsupervised tasks?

---

> ### Author Response · Authors · 2025-11-20
>
> We thank the reviewer for their positive assessment, for highlighting the innovation and biological grounding of PatchDNA. We are glad you found the re-patching flexibility and efficiency improvements to be practically useful contributions. We address your specific questions and suggestions below:
>
> >Can the authors conduct in-silico mutagenesis to confirm that patch boundaries capture causal biological signals?
>
> We agree that ISM is a valuable tool for interpretability of the model, and note it as a future research direction. We have decided to not apply ISM to PatchDNA due to time constraint, as setting up a systematic evaluation framework for this analysis is non-trivial. However, we address the core of your concern, i.e. verifying that patches capture biological signals through multiple analyses:
>
> * In **Table 3**, we demonstrate that when we use **cCRE-aware re-patching** (forcing patch boundaries to align with known causal regulatory elements), performance improves.
> * Similarly, in **Tables 4 and 5**, we show that using **DNase-seq signal** to define patches significantly improves cell-type specific predictions.
> * Additionally, we have included a new section **(A.8)** where we found that PhyloP-derived patches were significantly enriched within cCRE regions relative to randomly sampled non-cCRE genomic windows.
>
> >Can the authors report statistical significance for benchmark results and explore (or provide discussion on the potential of) PatchDNA’s performance on unsupervised tasks?
>
> We agree on the importance of reporting variance to ensure robust conclusions. We have reported standard deviations for all benchmarks where we managed the full training pipeline, specifically the Nucleotide Transformer Benchmark and CAGE prediction tasks.
>
> For DART-Eval and BEND, we strictly followed the official evaluation protocols provided by the benchmark authors for ensuring fair comparison. The standard reporting for these benchmarks (and the baseline results provided by the original authors) uses a single seed. To ensure our numbers were directly comparable to the published baselines, we matched this single-run setting. We will clarify in the final manuscript which benchmarks utilize multi-seed averaging and which follow single-seed protocols for consistency with baselines.
>
> Regarding PatchDNA’s performance on unsupervised tasks, our evaluation on DART-Eval Tasks 1, 2, and 5 is performed in a zero-shot setting. We use the model's likelihoods or raw embeddings without any supervised training or fine-tuning. The fact that PatchDNA achieves the best mean rank on DART-Eval (Table 1\) demonstrates strong unsupervised performance. The reviewers suggestion to assess PatchDNA on de novo sequence generation is noted as an exciting direction for future work as evaluation metrics for genomic generation mature.

---

### Official Review · Reviewer_gt3n · 2025-11-01

**Soundness:** 2
**Presentation:** 2
**Contribution:** 2
**Rating:** 4
**Confidence:** 4

**Summary:**

This paper introduces PatchDNA, a new framework for modeling DNA sequences that replaces traditional fixed tokenization methods with a flexible, dynamic "patching" mechanism. The core idea is inspired by the Byte Latent Transformer (BLT) and involves segmenting a DNA sequence into variable-length patches, where the patching strategy is biologically-informed.

**Strengths:**

1. The design of re-patching is biologically logical, which provides alternative tokenization to DNA modeling.
2. The paper provides extensive experimental results to support its claim.

**Weaknesses:**

1. Though the application in DNA domain seems to be successful, the methodological contributions are incremental to Byte Latent Transformer (BLT).
2. The additional experiments are needed to validate the efficiency of the proposed method.
2.1 The method's performance depends on a predefined threshold $θ_p$ to create patches. The paper adopts a fixed percentile, but a deeper analysis of how this choice impacts different tasks is missing. Although the appendix (Table 22) shows that performance varies with the threshold, indicating this is a non-trivial parameter to tune, a more comprehensive discussion is needed.
2.2 The current models are pre-trained exclusively on the human reference genome. While this is common practice, it limits the model's utility for genomics research on other species. Although Appendix A.5 shows promising zero-shot transfer to mouse, a truly foundational model for genomics should ideally be trained on a multi-species corpus to learn more universal biological principles.
2.3 There are missing baselines such as Evo and Evo2.
3. The model's performance is contributed by the inductive bias that evolutionary conservation is a universal proxy for functional importance. This reliance might become a weakness for tasks where functionally important regions are not conserved. When re-patching with a better signal is not an option (as in the case of a truly novel task), the model might be fundamentally disadvantaged by its pre-training, as the patching strategy itself could be misaligned with the task's biology.

**Questions:**

1. The choice of the patching threshold $θ_p$ seems critical. Could the authors provide more intuition or analysis on the trade-off it governs? For instance, how does the average number of patches and downstream performance change across a wider range of thresholds? Is there a risk of "over-patching" (losing too much resolution) or "under-patching" (losing efficiency gains) on certain tasks?
2. In Table 3, the cCRE-aware re-patching improves Gene and Cell Pearson scores but significantly degrades the Full Pearson score (from 0.471 to 0.408),  which is counter-intuitive.  Could the authors provide an explanation for this situation?
3. This work demonstrates remarkable parameter efficiency, which is a key goal in model design. Another critical property of successful architectures like the Transformer is their adherence to scaling laws. Have the authors investigated whether the PatchDNA framework exhibits predictable scaling behavior? Specifically, if the authors increase model parameters and training data, how does the method behave in contrast to standard tokenization methods?
4. In the context of building a general biological foundation model, we might encounter diverse or completely unknown input types, rendering task-specific re-patching impossible. In such a scenario, the choice of the scoring function $g_p$ becomes crucial. How would PatchDNA handle a task where the important regions are anti-correlated with evolutionary conservation? In such a scenario, does the framework offer a way to dynamically adapt the scoring function if a mismatch with the downstream task is detected?
5. The recently proposed Evo model is state-of-the-art on many genomic benchmarks. Although it is a much larger model, a comparison would be valuable. Have the authors considered benchmarking PatchDNA against Evo on any tasks?

---

> ### Author Response · Authors · 2025-11-20
> **Response (1/3)**
>
> We thank the reviewer for their feedback, and for highlighting the biological applicability and extensive experimental validation that we have performed. We respond to the points raised, below:
>
> >Though the application in DNA domain seems to be successful, the methodological contributions are incremental to Byte Latent Transformer (BLT).
>
> We would like to highlight that the novelty of our work lies in demonstrating the unique advantages of patching in DNA models. The BLT only investigates applications to NLP, and the extension that we have shown to DNA requires non-trivial methodological insights. We show that the lack of fixed vocabulary allows biologically relevant functions to be used in patching, going beyond the generic entropy based function used in BLT. We further introduce re-patching, by showing that this flexibility allows changing the patching post pretraining, bringing unique benefits to DNA modelling.
>
> >The choice of the patching threshold seems critical. Could the authors provide more intuition or analysis on the trade-off it governs? For instance, how does the average number of patches and downstream performance change across a wider range of thresholds? Is there a risk of "over-patching" (losing too much resolution) or "under-patching" (losing efficiency gains) on certain tasks?
>
> Thank you for raising this important point. In the paper, we choose the threshold such that the average patch size is 20, which allows PatchDNA to operate efficiently on long sequences (\>100 kbp) within our compute budget. We did not tune this during the main experiments. Since we show that the model at this patch size is highly competitive on a variety of downstream tasks, we recommend an average patch size of 20 as a default hyperparameter.
>
> To address the reviewer’s concern, we conducted additional experiments across a wide range of average patch sizes (4–80) on the full Nucleotide Transformer benchmark (18 tasks) and the CAGE long-range task. A detailed discussion and set of results are now included in Section A.7 in the supplementary of the revised manuscript.
>
> Based on these analyses, we find that smaller average patch sizes can offer additional benefit, but in most tasks the difference in performance is small. We also note that tokenizers in prior work have analogous hyperparameters (e.g., k value in k-mer tokenization), but modifying them requires pretraining an entirely new foundation model. In contrast, our framework allows users to re-patch at finer granularity without the cost of pretraining, enabling higher-resolution processing when required by a specific downstream task.
>
> >In Table 3, the cCRE-aware re-patching improves Gene and Cell Pearson scores but significantly degrades the Full Pearson score (from 0.471 to 0.408), which is counter-intuitive. Could the authors provide an explanation for this situation?
>
> Full Pearson is computed across all output bins, in which most values will be close to 0, as CAGE-seq is extremely sparse and localized to the TSS. Hence, this metric is dominated by how well the model identifies where the TSS is located within the sequence. We hypothesize that conservation-based signals are particularly effective for identifying the TSS, explaining why conservation based patching performs better on this metric. In contrast, Gene and Cell Pearson measure relative expression variation between genes or cell types, which are primarily driven by regulatory interactions, which the cCRE-aware strategy is designed to highlight.

---

> ### Author Response · Authors · 2025-11-20
> **Response (2/3)**
>
> >This work demonstrates remarkable parameter efficiency, which is a key goal in model design. Another critical property of successful architectures like the Transformer is their adherence to scaling laws. Have the authors investigated whether the PatchDNA framework exhibits predictable scaling behavior? Specifically, if the authors increase model parameters and training data, how does the method behave in contrast to standard tokenization methods?
>
> >The current models are pre-trained exclusively on the human reference genome. While this is common practice, it limits the model's utility for genomics research on other species. Although Appendix A.5 shows promising zero-shot transfer to mouse, a truly foundational model for genomics should ideally be trained on a multi-species corpus to learn more universal biological principles.
>
> We appreciate the reviewer’s perspective, multi-species training and scaling-law analyses are indeed important directions for building universal genomic foundation models. In this work, our focus was on evaluating the effectiveness of the patching framework in a controlled and widely used setting, which motivated pretraining on the human reference.
>
> Large-scale multi-species training is substantially more computationally intensive. For example, Evo2\[1\] uses \~2000 H100 GPUs \[2\], and the smallest multi species model we compare to, DNABERT-2\[3\] took 14 days on 8 NVIDIA RTX 2080Ti to train. Conducting scaling-law experiments would also be computationally expensive, requiring pretraining models at different parameter counts. Given these constraints, we prioritised demonstrating that patching is an efficient and high-performing approach across a broad range of downstream tasks; our evaluations span 36 tasks across four benchmarks. Notably, PatchDNA already exhibits competitive performance with substantially larger multi-species models such as Nucleotide Transformer despite being much smaller and trained on far less data. We view extending PatchDNA to multi-species genomes as a natural and exciting next step, and one that our framework is well positioned to support when the necessary compute resources become available.
>
> \[1\] Brixi, Garyk, et al. "Genome modeling and design across all domains of life with Evo 2." *BioRxiv* (2025): 2025-02.
>
> \[2\] https://blogs.nvidia.com/blog/evo-2-biomolecular-ai/
>
> \[3\] Zhou, Z., et al. (2024). DNABERT-2: Efficient Foundation Model and Benchmark for Multi-Species Genome. arXiv:2306.15006.
>
> >In the context of building a general biological foundation model, we might encounter diverse or completely unknown input types, rendering task-specific re-patching impossible. In such a scenario, the choice of the scoring function becomes crucial. How would PatchDNA handle a task where the important regions are anti-correlated with evolutionary conservation? In such a scenario, does the framework offer a way to dynamically adapt the scoring function if a mismatch with the downstream task is detected?
>
> Re-patching can precisely address this scenario. The framework does not rely exclusively on conservation signals: users can re-patch using fixed-size patches, entropy-based scores, or other heuristics. These alternatives remain available even when no task-specific annotations are present. If conservation is actively anti-correlated with a task, the scoring function can simply be adjusted, for example by inverting conservation scores or substituting a task agnostic signal such as fixed patching, allowing the model to adapt without incurring the cost of pretraining a new model.
>
> Importantly, in our empirical evaluation across a broad range of benchmarks, we do not observe cases where conservation-based patching significantly degrades performance. Some tasks exhibit minimal correlation with conservation, yet PatchDNA still performs competitively (see Tables 21, 22, 23 in Section A.4). [Edit: Table 22, 23, 24 in Section A.4 in latest version of the paper]

---

> ### Author Response · Authors · 2025-11-20
> **Response (3/3)**
>
> >There are missing baselines such as Evo and Evo2.
>
> >The recently proposed Evo model is state-of-the-art on many genomic benchmarks. Although it is a much larger model, a comparison would be valuable. Have the authors considered benchmarking PatchDNA against Evo on any tasks?
>
> We agree that the Evo family of models represents a significant milestone in genomic modeling and achieves state-of-the-art results. However, Evo is not trained on the human genome, so it is not directly comparable on our human-focused benchmarks.
>
> The recently published Evo2 model is trained on the human genome (as well as other species), so to address the reviewer’s concern, we show its published results on the zero-shot DART-Eval tasks (1, 2, and 5\) compared to our model. Due to compute constraints, we cannot run Evo2 on the remaining non-zero shot tasks ourselves.
>
> | Model | T1 Acc. | T2 Acc. | T5 AUROC (African) | T5 AUROC (Yoruban) |
> | :---- | :---- | :---- | :---- | :---- |
> | PatchDNA (20M) | 0.966 | **0.725** | 0.545 | 0.564 |
> | Evo2 (40B) | **0.976** | 0.633 | **0.660** | **0.580** |
>
> We would like to note that our model is roughly 2000× smaller than Evo 2, and trained on substantially less data.
>
> We do not aim to outperform models at this parameter scale, rather our goal is to demonstrate that patching provides an efficient and flexible architectural framework that allows a small model to perform competitively across diverse benchmarks. Even at this extreme size disparity, in many cases PatchDNA is comparable to Evo2 on the reported tasks.

---

### Official Review · Reviewer_rPgf · 2025-11-02

**Soundness:** 3
**Presentation:** 3
**Contribution:** 3
**Rating:** 6
**Confidence:** 4

**Summary:**

This paper introduces PatchDNA, which draws on the ideas of the Byte Latent Transformer (BLT) to shift DNA sequence modeling from traditional tokenization to "patching." The authors further introduce patch boundary selection based on conservation signals and a "re-patching" mechanism that allows for the redefinition of patch strategies after pre-training, thereby enhancing flexibility and efficiency.

Core Contribution: By leveraging concepts from computer vision (CV) and BLT, the approach transforms the tokenization problem into a "patching" problem, completely eliminating the need for a fixed vocabulary. It utilizes external biological signals, such as conservation scores (PhyloP), to guide the delineation of patch boundaries.

**Strengths:**

- The idea is novel, completely eliminating the limitations of a fixed vocabulary and circumventing the rigidity of k-mers and the statistical constraints of BPE.
- By guiding patch boundaries with conservation scores, it provides a biological inductive bias.
- Experimental results show performance that is comparable to or even surpasses that of large-scale models, while being more computationally efficient.

**Weaknesses:**

- Non-End-to-End: The patch boundaries rely entirely on external signals (conservation, entropy), and the model itself does not learn the importance of the regions.

- Insufficient Empirical Support: The performance improvements from re-patching are not adequately quantified, remaining largely at the conceptual level.

- Limited Interpretability: There is a lack of in-depth analysis of the relationship between the representations learned by the model and biological functions.

- High Implementation Complexity: While being "vocabulary-free" offers flexibility, it results in poor standardization and reproducibility.

**Improvement Suggestions:**

- Introduce a trainable boundary scoring network to jointly optimize patch segmentation and representation learning.
- Conduct a quantitative analysis of the actual impact of re-patching on performance and efficiency.
- Increase the analysis of the overlap between patches and functional regions (conserved regions, regulatory elements).
- Publicly disclose memory usage and computational load comparisons to ensure the claimed efficiency is credible.


Besides, this paper seems to have been previously presented at a NeurIPS Workshop (https://openreview.net/group?id=NeurIPS.cc/2025/Workshop/AI4D3&referrer=%5BHomepage%5D(%2F)#tab-accept-oral); however, my comments do not take this into consideration. I leave it to the AC to make a judgment on this matter.

**Questions:**

please refer to the weaknesses part

---

> ### Author Response · Authors · 2025-11-20
>
> We thank the reviewer for recognizing the novelty of our vocabulary-free approach and our model’s ability to surpass large-scale baselines with greater efficiency. We appreciate the constructive feedback and address your specific concerns below.
>
> >Non-End-to-End: The patch boundaries rely entirely on external signals (conservation, entropy), and the model itself does not learn the importance of the regions.
>
> >Introduce a trainable boundary scoring network to jointly optimize patch segmentation and representation learning.
>
> We appreciate this suggestion and agree that a learnable segmentation module is an exciting direction for future work.
> In this work, however, our goal is to introduce patching as a practical and efficient framework for DNA representation learning. Designing and training an end-end boundary network for sequences exceeding 100 kbp would also add substantial implementation and computational complexity.
>
> >Insufficient Empirical Support: The performance improvements from re-patching are not adequately quantified, remaining largely at the conceptual level.
> >Conduct a quantitative analysis of the actual impact of re-patching on performance and efficiency.
>
> We would like to highlight that the manuscript already contains several quantitative demonstrations of how re-patching affects performance.
>
> First, on the long-range CAGE benchmark, using cCRE-aware patching only during fine-tuning improves Gene Pearson from 0.369 → 0.373 and Cell Pearson from 0.771 → 0.792 for PatchDNA-7M, without retraining from scratch (Table 3). Additionally, DNase-aware re-patching further improves cell-wise Pearson from 0.754 → 0.828 (K562), 0.717 → 0.727 (hepatocytes), and 0.799 → 0.831 (neurons) over the same base model (Tables 4 and 5). Second, in Table 12, in a particular task, we show that re‑patching a model pretrained with conservation-based patching to single-nucleotide patches can yield substantial gains, but without paying the large pretraining cost at that resolution.
>
> To further strengthen this point, in the revised manuscript (Section A.7), we have added a systematic, quantitative analysis of re-patching and patch-size / threshold sensitivity across multiple benchmarks. These experiments directly quantify how re-patching changes both performance and effective resolution without pretraining new models. Importantly, all these results are obtained from a single pretrained model by changing only the patching function, illustrating that re-patching provides a practical approach to trade off granularity and efficiency without retraining a new model at each resolution.
>
> >Increase the analysis of the overlap between patches and functional regions (conserved regions, regulatory elements).
>
> We agree that it is important to directly analyze how patch boundaries relate to known functional annotations. As such, we have included a new section A.8 where we conducted further analyses and found that there were significantly more PhyloP-derived patches within cCRE regions relative to randomly sampled non-cCRE genomic windows.
>
> >Publicly disclose memory usage and computational load comparisons to ensure the claimed efficiency is credible.
>
> We appreciate this suggestion to further substantiate our efficiency claims. Table 18 in section A.3.4 reports FLOPs, and we have now updated it to explicitly include Peak VRAM usage (GB) during fine-tuning. This comparison confirms that PatchDNA requires less memory than leading long-context DNA baselines while maintaining competitive performance.
>
> >High Implementation Complexity: While being "vocabulary-free" offers flexibility, it results in poor standardization and reproducibility.
>
> Thank you for raising this concern. We will release full code upon publication, including implementations of patching, our conservation-based scoring functions, and re-patching. Additionally, section A.6 includes explicit pseudocode for patch-boundary detection and a minimal code example showing how re-patching is done in practice.

---

> > ### Comment · Reviewer_rPgf · 2025-11-25
> > **Final Rating**
> >
> > Thanks for the responses, and most of my concerns are addressed. Besides, I also read the comments from other reviewers.
> > Although some issues exist, I believe this work is beneficial to the community. I tend to accept this paper and maintain my initial rating (6).

---

### Author Response · Authors · 2025-11-20
**General response to all reviewers**

We thank reviewers for their reviews and for highlighting the key strengths of our paper, including its strong biological grounding, efficiency, and the practical flexibility introduced by re-patching.

We also thank the reviewers for their suggestions to improve the manuscript, and we present a revised version, with new additions in blue. We detail the new additions below:

**Average patch size analysis and discussion**
Based on feedback from Reviewer gt3n and Reviewer 3uWp, we have expanded Section A.7 in supplementary to include a detailed analysis of model performance at a wide range of average patch sizes and long-range and short-range downstream tasks.

**Extended results for BEND and Genomics Long Range Benchmark**
Based on feedback from Reviewer 3uWp, we have included extra tasks from the BEND and Genomics Long Range Benchmark, which cover extra **long range tasks,** and **variant effect prediction.**

**Biological interpretability of patches**
Based on feedback from Reviewer rPgf, we add a section performing a quantitative analysis of patch alignment with regulatory regions (Section A.8).

**Memory usage reporting**
Based on feedback from Reviewer rPgf, we have added a column to Table 18 (Section A.3.4) reporting the peak VRAM usage (GB) during full model finetuning to supplement the previously reported wall-clock time and FLOPs.

We also address each reviewer's concerns individually in separate comments.

---

### Author Response · Authors · 2025-12-03
**Summary of rebuttal and discussion**

We thank the reviewers for their feedback and discussion, which has further strengthened the paper. To summarise, we believe that our rebuttal directly addresses the main reviewer concerns, with substantial new experiments and clarifications. **We highlight that the feedback we received from our rebuttal was positive**, where reviewer 3uWp who gave the lowest score indicated that they were satisfied with our response and would increase their score.

In addition to the changes we mentioned in our general comment, we have also added the full set of Genomics Long Range Benchmark tasks to the paper, showing that our model outperforms other long range DNA language models on 6 out of 7 tasks.

We also thank the reviewers for highlighting the strengths of our paper, particularly its biological grounding, empirical breadth, efficiency, and the practical utility of re-patching. We believe that patching in DNA as presented by our paper, offers important unique advantages for DNA modeling (efficient long-sequence processing, single-nucleotide resolution, and the ability to incorporate biologically meaningful inductive biases) providing a strong foundation for future work in the field.

---

### Meta-Review · Area_Chair_gVg6 · 2026-01-04

**Summary:**

This paper proposes PatchDNA, a patch-based alternative to fixed tokenization for DNA sequence modeling, with biologically informed boundary selection and a flexible re-patching mechanism that allows changing segmentation after pretraining. Overall, the reviewers agree that the approach is well motivated and that the empirical evaluation is strong.

The rebuttal addressed most of the concrete concerns raised during review. In particular, in response to gt3n and 3uWp, the authors added a systematic analysis of patch-size sensitivity and substantially expanded long-range evaluations (BEND and the Genomics Long Range Benchmark). Requests from rPgf regarding interpretability and efficiency were also addressed through new analyses of patch alignment with regulatory regions and explicit reporting of memory usage. The main empirical gaps identified during review were largely resolved, and the most critical reviewer (3uWp) indicated that several of their earlier concerns had been addressed.

Some limitations remain. gt3n’s concern that the methodological contribution may be incremental relative to BLT-style patching, and rPgf’s point that the framework relies on externally defined patching signals rather than an end-to-end learned segmentation, are not fully resolved. In addition, the paper does not yet provide a deep mechanistic explanation of why re-patching remains robust under the distribution shift it induces. These issues explain why a reject decision would also have been defensible. That said, given the strength and breadth of the empirical results and the constructive rebuttal, I lean toward acceptance (poster).

**Reviewer Concerns:**

**Reviewer gt3n**

This reviewer raised concerns about the sensitivity of the method to the patching threshold and the lack of analysis on how average patch size affects downstream performance. The rebuttal added a systematic patch-size sensitivity study across a wide range of average patch sizes and expanded the discussion accordingly, addressing this concern.

The reviewer’s concern that the methodological contribution may be incremental relative to BLT-style patching remains a conceptual issue and was not fully resolved by additional experiments.

**Reviewer 3uWp**

This reviewer was initially concerned about insufficient long-range evaluation, limited coverage of variant-level tasks, and unclear empirical support for some claims. The rebuttal substantially expanded the experimental coverage by adding the full BEND benchmark and the Genomics Long Range Benchmark, as well as additional variant effect prediction results and ablations. Following these additions, the reviewer indicated that several key concerns were addressed and increased their score.

While most empirical concerns were addressed, questions related to deeper mechanistic understanding (e.g., why re-patching remains robust under distribution shift) were discussed but not fully resolved.

**Reviewer rPgf**
This reviewer requested stronger empirical support for efficiency and interpretability claims, including explicit reporting of memory usage and analysis of the biological relevance of patch boundaries. The rebuttal added peak VRAM usage reporting and a quantitative analysis showing enrichment of patch boundaries within regulatory regions, addressing these concerns.

The reviewer’s concern that the approach is not end-to-end and relies on externally defined patching signals remains an inherent limitation of the current framework.

**Reviewer 6uT1**
This reviewer suggested adding statistical reporting and clarifying evaluation paradigms. The rebuttal clarified which benchmarks follow multi-seed versus single-seed evaluation protocols and expanded the discussion of zero-shot and unsupervised evaluations.

The reviewer suggested additional mechanistic or causal analyses (e.g., in-silico mutagenesis) to further validate biological claims. These were acknowledged but left as future work.

**Why accept**
Despite initial score divergence, the rebuttal addressed most concrete empirical concerns raised during review, particularly regarding long-range evaluation, patch-size sensitivity, and efficiency. The remaining issues primarily concern conceptual framing and deeper mechanistic understanding, rather than correctness or empirical validity. Given the strength and breadth of the experimental results and the constructive trajectory of the discussion, a poster accept is warranted.

**Reviewer Scores:**

- Reviewer 6uT1: Likely to maintain a high score (≈8).
- Reviewer rPgf: Likely to maintain their original score (≈6).
- Reviewer gt3n: Likely to remain borderline (≈4–5)
- Reviewer 3uWp: Likely to increase from reject-level to borderline/weak accept (≈4–5)

---

### Decision · Program_Chairs · 2026-01-26

Accept (Poster)